# Changes of the Arctic marginal ice zone during the satellite era

Rebecca J. Rolph*, Daniel L. Feltham, David Schröder

Centre for Polar Observation and Modelling, Department of Meteorology, University of Reading, Reading, RG6 6BB, UK
* now at Alfred Wegener Institute Helmholtz Centre for Polar and Marine Research, Potsdam, Germany

5 *Correspondence to*: Rebecca J. Rolph (rebecca.rolph@awi.de)

**Abstract.** Many studies have shown a decrease in Arctic sea ice extent. It does not logically follow, however, that the extent of the marginal ice zone (MIZ), here defined as the area of the ocean with ice concentrations from 15 to 80%, is also changing. Changes in the MIZ extent has implications for the level of atmospheric and ocean heat and gas exchange in the area of partially ice-covered ocean, as well as for the extent of habitat for organisms that rely on the MIZ, from primary producers like sea ice 10 algae to seals and birds. Here, we present, for the first time, an analysis of satellite observations of pan-Arctic averaged MIZ extent. We find no trend in the MIZ extent during the last 40 years from observations. Our results indicate that the constancy of the MIZ extent is the result of an observed increase in width of the MIZ being compensated by a decrease in the perimeter of the MIZ as it moves further north. We present simulations from a coupled sea ice-ocean mixed layer model using a prognostic floe size distribution which we find is consistent with, but poorly constrained by, existing satellite observations of 15 pan-Arctic MIZ extent. We provide seasonal upper and lower bounds on MIZ extent based on the 4 satellite-derived sea ice concentration datasets used. We find a large and significant increase (>50%) in the August and September MIZ fraction (MIZ extent divided by sea ice extent) for the Bootstrap and OSI-450 observational datasets, which can be attributed to the reduction in total sea ice extent. Given the results of this study, we suggest that references to 'rapid changes' in the MIZ should remain cautious and provide a specific and clear definition of both the MIZ itself and also the property of the MIZ that is changing.

## 1 Introduction

Arctic sea ice extent has been declining rapidly during the last 40 years (Comiso et al., 2008; Onarheim et al., 2018; Serreze et al., 2007; Stroeve et al., 2007). The MIZ has been variously defined as where ocean wind-generated waves interact with the sea ice (e.g. Dumont et al., 2011) or as the area of ocean covered with 15-80% sea ice (e.g. Aksenov et al., 2017; Strong and Rigor, 2013). Due to its utility and the wealth of sea ice concentration data available, we use the latter operational definition 25 of the MIZ extent being the total area of ocean capped by 15-80% sea ice cover. Given the rapid decline of sea ice extent in the Arctic, associated studies consequently tend to assume that the marginal ice zone (MIZ) is expanding (Boutin et al., 2020; Lee and Thomson, 2017; Strong et al., 2017; Horvat and Tziperman, 2015). The purpose of this paper is to show whether the *extent* of MIZ, defined in this study according to the operational characterization, is actually changing. While there are significant regional changes happening in the Arctic MIZ such as increased light penetration (e.g. PAR), open water, and gas 30 exchange (Barber et al., 2015), it is important to keep in mind that these changes are not necessarily a result of a change in the

coverage of the total MIZ, but rather more likely the change in its location. As the Arctic MIZ moves northwards (Aksenov et al., 2017), the increased southward area of open ocean subsequently allows for increased wind-wave generation which can break up the ice (Collins et al., 2015; Thomson and Rogers, 2014). Thinner ice cover (Kwok, 2018) in combination with an increase in wind-wave action may result in smaller floes that melt faster due to an increased lateral melt rate (Steele, 1992;

Tsamados et al., 2015). The MIZ can also contribute to Arctic amplification because it is an area for Arctic cyclogenesis which is important for northward meridional heat transport (Inoue and Hori, 2011). The MIZ supports many important processes such as Arctic marine primary production (Alexander and Niebauer, 1981), delivery of nutrients to the euphotic zone, air-sea gas exchange, and carbon exchange across the air-sea interface (Barber et al., 2015; Hansen et al., 1996). Monitoring changes of the MIZ environment in which these processes occur can help us understand the associated changes in the climate system.


It has been found that the width of the MIZ has increased in the summer and decreased in winter from 1979-2011 (Aksenov et al., 2017; Strong and Rigor, 2013). However, it was also found that there is a positive (northward) trend in the area-weighted latitude of the MIZ during the same time period (Strong and Rigor, 2013). A northward trend of the MIZ and an increase of its (summer) width does not necessarily imply that the MIZ extent is increasing as the effective perimeter of the MIZ may be

decreasing. A decrease of total sea ice extent combined with a widening of the MIZ does imply, however, that the central pack ice will occupy less area. This could ease Arctic access for ships (Aksenov et al., 2017). While the Arctic is projected to have entirely seasonal ice cover by mid-century (Notz and Stroeve, 2018; IPCC, 2014), a study of specific trends in MIZ extent is lacking, such as quantification of the MIZ extent relative to the total sea ice extent. Thus, we need to remain cautious and provide a specific and clear definition of the property of the MIZ when stating the Arctic MIZ is 'rapidly changing'. It also

follows that we need to be aware of the extent to which our observations are able to constrain any model of the MIZ. Note that Stroeve et al. (2016) have examined MIZ extent in the Southern Ocean.

We use a state-of-the-art sea ice-ocean model to better understand how well simulations can capture the satellite-observed MIZ. Due to the nature of the operational MIZ extent definition used here, this study can also be viewed as a test of model

performance concerning how well sea ice concentrations are simulated on a pan-Arctic scale. Winter, summer, and autumn months were selected to illuminate how well observations and simulation agree on a seasonal timescale. The bulk of the model set-up follows Schröder et al. (2019) and can be seen as representative of how well other models simulate sea ice concentration.

The paper set-up is as follows. In Section 2, we introduce the satellite observational datasets and model set-up we used. In

Section 3, we describe the methods of applying our satellite data to our model grid, and subsequent analysis of the results. Here we also describe how we defined the MIZ and sea ice cover in our calculations. Section 4 presents our analysis of the extent of the total sea ice cover and MIZ as monthly averages for March, July, August, and September for the period from 1979-2017. It also includes trends and statistical analysis of the total MIZ extent relative to the total ice extent. Section 4

discusses the apparent change of location in the MIZ. The subsequent discussion (Section 5) outlines possible implications of the trends we observe, and what this could mean for future projections of the MIZ.

## 2 Model set-up and Data

### 2.1 Observational datasets

The satellite products used in this study are: OSI-450 (EUMETSAT), NASA Bootstrap (Comiso, 2017), AMSR-E and AMSR-2 (Spreen et al., 2008). OSI-450 is the second version of a processing of the EUMETSAT Ocean and Sea Ice Satellite Application Facility (OSI SAF) team Sea-Ice Concentration (SIC) Climate Data Record (CDR). Sea ice concentration is provided over the polar regions at 25 km resolution and derived from passive microwave satellite data SSMR, SSM/I and SSMIS for years 1979 through 2015 (every other day from January 1, 1979 to August 20, 1987 and daily from August 21, 1987 through December 31, 2015). This processing includes using Numerical Weather Prediction re-analysis atmospheric data to correct brightness temperature, dynamic tie-points, and state-of-the-art algorithms which are described in detail in (Lavergne et al., 2016). The NASA Bootstrap sea ice concentration product has a 25 km resolution and is derived from SSMR, SSM/I, and SSMIS sensors and generated using the AMSR-E Bootstrap Algorithm (Comiso, 2017) with daily data available from November 1978 through 2018. The AMSR-E Bootstrap algorithm uses daily varying tie-points, three frequency channels which are available continuously from SSMR, through SSM/I and to AMSR-E. All three of these channels have vertical polarization and two of those have horizontal polarization. A basic assumption of the Bootstrap algorithm is that a certain observational area is covered by entirely ice or water, which can lead to data smearing at the ice-ocean edge or in areas where the contrast of emissivity between ice and water are not so strong. A higher resolution in general gives better chances to distinguish the correct location of the ice edge and characterize the MIZ (Comiso, 2012).

AMSR-E v5 and AMSR-2 v5.4 are datasets processed using the ASI-algorithm (Spreen et al., 2008) and are the highest resolution observational datasets used in this study with a grid spacing of 6.25 km. The time frame for available data for the Japanese AMSR-E sensor onboard the EOS/Aqua satellite is from 1 June 2002 to 4 October 2011 and for the AMSR-2 sensor onboard the GCOM-W satellite is from July 2012 to 17 November 2018. The ice concentration is calculated from the difference of brightness temperatures in the vertical and horizontal polarization, which is a result of emissivity differences. At 90 GHz, the emissivity of open water is much smaller than that of all ice types and so water can be distinguished from the ice. An atmospheric correction is applied to account for the influence of the atmosphere on the upwelling polarization (Spreen et al., 2008). This correction assumes a horizontally stratified Arctic atmosphere with an effective temperature to replace the vertical temperature profile and a diffusely reflecting surface viewed under a 50º incidence angle (Svendsen et al., 1987). The ice concentration then becomes a function of the polarization difference and the atmospheric correction term which is in general also a function of ice concentration (Svendsen, 1983; Svendsen et al., 1987). Atmospheric influence is assumed to be a smooth function of ice concentration and a third order polynomial for ice concentration is solved as a function of polarization

difference. Fixed tie points, which provide necessary values for unknowns, are found by comparing ice concentration from the Svendsen algorithm and an ice concentration reference from an independent source that has been well validated (Spreen et al., 2008). A weather filter is applied due to the disadvantage of the brightness temperatures from 89 GHz channels being influenced by the atmospheric cloud liquid water and water vapor. Some sources of error include water vapor and wind roughening of the ocean influencing the polarization difference. Values for error between the different data products used in this study are given in Section 5.5.

## 2.2 Model set-up (CICE-CPOM-2019)

We use a dynamic-thermodynamic sea ice model, CICE-CPOM-2019, which is designed to be included in global climate models. CICE-CPOM-2019 is based on the existing CICE model version 5.1.2, but with some additions. We perform a stand-alone (fully forced) simulation for the pan-Arctic region (~40 km grid resolution) with a spin-up of 10 years from 1979, and then restarted at 1979 and run through 2016. The CICE model solves 1-D vertical heat balance equations for 5 ice thickness categories. The momentum balance equation which provides the sea ice velocity includes air and ocean drag, the Coriolis force, sea surface tilt, and internal ice stresses. Hunke et al., (2015) gives a detailed description of the CICE model. Since we did not use a coupled ocean model to calculate heat transport in the ocean or ocean currents, the temperature and salinity in the layer below the ocean mixed layer are restored every 20 days to climatological monthly means from MYO-WP4-PUM-GLOBAL-REANALYSIS-PHYS-001-004 (Ferry et al., 2011). Ocean currents are restored on the same timescale and from the same reanalysis dataset. For the atmospheric forcing, NCEP Reanalysis-2 (NCEP2) is used (Kanamitsu et al., 2002, updated 2017).

Some of the default CICE configurations used in this study include: seven vertical ice layers, one snow layer, thermodynamics of Bitz and Lipscomb (1999), Maykut and Untersteiner (1971) conductivity, the Rothrock (1975) ridging scheme with a Cf value of 12 (an empirical parameter that accounts for dissipation of frictional energy), the delta-Eddington radiation scheme (Briegleb and Light, 2007), and linear remapping ITD approximation (Lipscomb and Hunke, 2004). For CICE-CPOM-2019, we switched on a prognostic melt pond model (Flocco et al., 2010, 2012), used an elastic anisotropic plastic rheology (Heorton et al., 2018; Tsamados et al., 2014; Wilchinsky and Feltham, 2006), a prognostic oceanic mixed layer (Petty et al., 2014) and a prognostic floe size distribution (Roach et al., 2018). Demonstrated use of CICE-CPOM-2019 including the above additions, with the exception of the prognostic mixed layer and floe size distribution, is provided as the reference simulation in Schröder et al. (2019). The prognostic mixed layer allows the ocean below the mixed layer to be relaxed toward observations so that the mixed layer can calculate its salinity, temperature, and depth based on the fluxes from the deeper ocean (Petty et al., 2014). The prognostic floe size distribution is a new development (Roach et al., 2018) and warrants more detailed description which is provided in the next section.

### 2.2.1 Prognostic floe size distribution

A sea ice floe size distribution is a probability distribution function (Thorndike et al., 1975) that characterizes the extensive variability (centimeters to hundreds of kilometers) in the range of sea ice floe sizes. Imposing a subgrid-scale floe size distribution (e.g. Bennetts et al., 2017; Zhang et al., 2016) does not account for physical processes acting on individual floes. However, here we have added the recent development by Roach et al. (2018) into CICE-CPOM-2019, which accounts for ice formation, welding of floes, lateral freeze/melt, and fracture by ocean surface waves. Particularly important processes which determine the floe size evolution are lateral melt of floes and floes welding together, as well as wave fracture. When floes are smaller, the lateral melt becomes more important, and this can lead to a significant reduction in sea ice concentration in summer (Roach et al., 2018). CICE simulates an ice thickness distribution and the sea ice concentration is calculated by integrating over all ice thickness categories. The change in the ice thickness distribution depends on growth/melt at a melting/freezing rate, ice advection, and redistribution of thickness categories caused by sea ice deformation. When the heat available from the surface of the ocean is enough to melt the ice, basal melting will occur by balancing the conductive heat flux from the bottom and downward heat flux from the ice to the ocean. Lateral melt is obtained as a function of floe size. CICE uses a constant floe size of 300 m, but in CICE-CPOM-2019 a joint floe-size thickness distribution (FSTD) is used that has been implemented and developed by Roach et al. (2018) following Horvat and Tziperman (2015).

The thermodynamic changes in the FSTD not included in the standard CICE model include a welding parameter for newly formed floes to freeze together and a 'lead region' which is part of the open water fraction where lateral growth of existing floes can occur around non-circular floes. Mechanical breaking of sea ice floes by ocean surface waves is determined by a critical strain and minimum floe size (10 m) which can be impacted by wave fracture. The fractures that would occur if the waves enter an entirely ice-covered region defined in the 1-dimensional direction of propagation are calculated and then the outcome is applied proportionally to each grid cell's ice-covered fraction. Swell from hindcast wave data coming from the equatorward meridional direction are used to select the significant wave height and mean period. This is then used to construct the wave spectrum (Horvat and Tziperman, 2015) which is attenuated exponentially given the number of floes in the grid cell, and is a function of ice thickness and wave period. With the assumption that sea ice flexes with the sea surface height field, the strain of the ice is calculated and the floe will fracture if this crosses a threshold. New floe radii are put into a histogram which depend on the local sea surface height field only.

### 3 Methods

#### 3.1 Applying satellite-derived sea ice data to model grid

All available OSI-450, NASA Bootstrap, and AMSR data through 2017 (2015 for OSI-450) were interpolated onto the ORCA tripolar 1° grid. A tripolar grid allows a construct of a global orthogonal curvilinear ocean mesh that has no singularity point

inside the computational domain because two north mesh poles are placed on land (Madec and Imbard, 1996). The ORCA tripolar grid is used by CICE and so will hereafter be referred to as the 'CICE grid' for simplicity. We use about a 40 km resolution mesh, with the CICE land mask also applied. For NASA Bootstrap, AMSR-E, and AMSR2, a data gap at the pole exists in the downloaded product we filled in. To do this, after interpolating the daily satellite data to the CICE grid, we marked which grid cells at the pole were missing sea ice concentration data. Then, we re-gridded each daily file onto a lower-resolution grid such that the missing values near the pole could no longer be resolved. We then applied this output back to the original higher resolution CICE grid. However, only the values of those grid cells which had previously been missing data on the CICE grid were kept from this method. One exception to this pole-filling method includes the years of 1979 through 1987 in the Bootstrap data, where the pole gap was larger than the rest of the Bootstrap data and the interpolation to the coarser grid still resolved some of the pole gap. Based on the high surrounding summer ice concentration (>80%) for these early years, the sea ice concentration within the pole gap is expected to be over 80%, so this was assumed for these years. The rest of the values in the CICE grid were taken via direct interpolation of the satellite data.

## 3.2 Calculating the marginal ice zone and sea ice extent

The MIZ extent was calculated as the total area of all grid cells between the thresholds of 15% and 80% sea ice cover, as the MIZ is also defined by other studies (e.g. Strong and Rigor (2013), Aksenov et al., (2017)). The daily values of MIZ extent were calculated for each of the observational datasets after they had been re-gridded to the model grid (and model land mask applied). The daily values of model MIZ extent were calculated from the model output. The sea ice area within the MIZ was also calculated for all observational datasets in this study (Bootstrap, OSI-450, AMSR-E, AMSR-2) as well as the model. The daily total sea ice extent was found for each dataset, which is defined as the total area of those grid cells which are covered by at least 15% sea ice. The daily MIZ extent was divided by the daily sea ice extent to get the daily relative MIZ extent. The monthly means of all these daily metrics were then calculated, and the main further analysis has used these monthly means. AMSR-E and AMSR-2 were combined into one time series, labelled AMSR, for the purpose of cross-correlating with the other datasets. We were unable to derive the error associated with these total measures of extent from the satellite products themselves due to uncertainties in the processing chains that prevent clear statements of error bounds. Following Spreen et al. (2008), we apply an error of 10%, based on systematic differences of monthly satellite products, to our calculated monthly means of the sea ice extent, MIZ extent, as well as the relative MIZ extent. The $r^2$ values are calculated using a linear least-squares regression and alpha represents a 95% confidence level.

## 3.3 Approximating changes in MIZ geometry

We next investigated how the changes in MIZ width and position (latitude) impact its extent. The monthly means of the latitudes of all MIZ grid cells were quantified for Bootstrap, OSI-450, and CICE-CPOM-2019. The timeseries of the latitudes for the MIZ found in the AMSR datasets was not calculated due to the relatively shorter temporal coverage compared to the other datasets. The trendlines of the yearly timeseries of the monthly MIZ latitude means were calculated with the associated

RMS values. The radius of the MIZ was approximated by $R_{MIZ} = R_{Earth} * \cos(\Theta_{MIZ})$ where $\Theta_{MIZ}$ is the monthly-averaged MIZ latitude and $R_{Earth}$ is the radius of the earth. The radius of the MIZ as described here refers to the radius of the parallel on which the MIZ is centered (measured perpendicular to the Earth's axis of rotation). The MIZ outer perimeter ($P_{MIZ}$), or the circumference of the parallel on which the MIZ is centered, was then approximated from the average latitude of all MIZ grid cells while assuming a spherical earth and no land. This was done by substituting $R_{MIZ}$ for the radius in the perimeter equation for a circle: $P_{MIZ} = 2*\pi*R_{Earth} * \cos(\Theta_{MIZ})$. Because we assumed no land when calculating the average perimeter of the MIZ, we focused on the summer months when the ice is, in general, north of the main northern hemisphere landmass. Since we had previously found the extent of the MIZ (Section 3.2), the MIZ width could be approximated using the simple formula: MIZ Width = MIZ Extent / MIZ Perimeter. For July, August, and September, the change in MIZ width and MIZ perimeter with associated RMS values were calculated from the slope of each yearly timeseries, while setting the change in MIZ extent to zero. The fraction of the MIZ extent that must be reduced as the MIZ trends northward, given no change in width, was approximated using Equation 1 below where the initial and final values for each variable are taken from the trendlines of the respective yearly timeseries of each July, August, and September month.

$$\frac{P_{MIZ\,(final)}}{P_{MIZ\,(initial)}} = \frac{2\pi\,R_{MIZ\,(final)}}{2\pi\,R_{MIZ\,(initial)}} = \frac{2\pi\,R_{Earth}*\cos(\theta_{final})}{2\pi\,R_{Earth}*\cos(\theta_{initial})} = \frac{\cos(\theta_{final})}{\cos(\theta_{initial})} \quad \text{(Eq. 1)}$$

Equation 1 gives the fraction of the MIZ extent which has decreased due to the decreased perimeter caused by the MIZ moving northwards. The inverse of Equation 1 was calculated to find the fraction of the MIZ width that must increase for the extent to remain constant. The fractions that the MIZ width must increase for the extent to remain constant using the approximation as given in Equation 1 are compared with the fraction change of the MIZ width found from the trends of the average latitudes of MIZ grid cells in the Bootstrap, OSI-450, and CICE-CPOM-2019 products.

## 4 Results

### 4.1 Extent of marginal ice zone and total sea ice

The sea ice extent across the observational products do agree within their range of uncertainty. The model simulation agrees with the observations during winter, but slightly underestimates the summer ice extent (solid lines in Fig. 1). The sea ice extent as calculated by the model still falls within the error range through July (solid lines in Fig. 1b) and is underestimated starting in August (solid lines in Fig. 1c) and September (solid lines in Fig. 1d). However, by October, the ice extent is again within the 10% error range within the observational products. The March, July, August and September trends of declining total sea ice extent (Table 1) are significant with the exception of the modelled trend in the March sea ice extent. September shows the fastest rate of decline compared to the other months examined, consistent with other studies (Boé et al., 2009).

There is also a significant high correlation between the inter-annual variability of sea ice extent observations for all months examined, with values greater than 0.957 in March and greater than 0.987 for July, August and September (Table 2). The lowest correlations occur in March between the model and OSI-450 (0.448), the model and Bootstrap (0.587), and also in July between the model and AMSR (0.575).

In contrast with the sea ice extent, there is no significant trend in the MIZ extent in any of the observational datasets, with the exception of a small negative trend in Bootstrap in March of -0.52% or -0.520 x $10^4$ km$^2$ per year (Fig. 1, Table 1). There is also no significant trend in the modelled MIZ extent except for September (roughly 1.1% or -1.37x$10^4$ km$^2$, r$^2$ = 0.31, Fig. 1, Table 1). For most of the summer months, the spread of observations of MIZ extent is greater than the 10% error placed on each of the observations themselves (Fig. 1b-d). This indicates that the observational error for the MIZ is larger than our 230 assumed value of 10% based on Spreen et al. (2008). The modelled MIZ extent generally lies within the spread of the observations. The observations taken together provide lower and upper bounds for MIZ extent of between roughly 5-15 x$10^5$ km$^2$ for March, 15-50 x$10^5$ km$^2$ for July, 15-45 x$10^5$ km$^2$ for August, and 10-30 x$10^5$ km$^2$ for September (Fig. 1).

The spread of the MIZ is larger than the sea ice extent in the observations (Fig. 1). In the winter months (e.g. dashed lines in 235 Fig. 1a), the MIZ extent is more consistent across the datasets. In March, there are significant correlations between the MIZ extent observations (>0.889, Table 2) as well as for the model results. From July through August, the differences in the absolute MIZ extent become very pronounced (dashed lines in Fig. 1b-d). In July, the AMSR and Bootstrap are the most highly correlated (0.869), with lower or insignificant values between the other datasets. In September, the AMSR is well correlated with the other observations of MIZ extent (0.805 with OSI-450 and 0.852 with Bootstrap, Table 1).

**4.2 Fraction of MIZ relative to total sea ice extent**

The trends for the MIZ fraction, i.e. MIZ extent divided by the sea ice extent, for all of the observations are insignificant for March, but slightly positive for July, August, and September with the exception of AMSR, which is insignificant for July and September (Fig. 2, Table 1). The trends per year for July are +0.003 for OSI-450 and +0.002 for Bootstrap. In August, there is an increase in MIZ fraction per year of 0.005 for OSI-450, 0.003 for Bootstrap, and 0.008 for AMSR. In September, the 245 positive significant trends per year are 0.004 for OSI-450 and 0.003 for Bootstrap. The positive trend in MIZ fraction is given by the stable MIZ extent and decline in sea ice extent (compare dashed and solid lines in Fig. 1). The MIZ fraction for OSI-450 is consistently higher compared to the other observational datasets (Fig. 2). The Bootstrap MIZ extent (absolute) is lower than OSI-450, which is the main reason for its lower MIZ fraction. The MIZ fraction for the model is insignificant for March, but slightly positive for July, August and September at +0.009, +0.010 and +0.003 per year, respectively. In July, CICE-250 CPOM-2019 model shows a trend in MIZ fraction three times that of the OSI-450 and over four times that of the Bootstrap dataset. In August, the model shows a trend two times that of OSI-450 and over three times that of Bootstrap. In September,

the trends of MIZ fraction become roughly the same in the model and observations, and this remains so during the winter months (Table 1).

The modelled MIZ fraction generally lies within the spread of the observations with the exception of August, where it is overestimated (Fig. 2). The observations taken together provide lower and upper bounds for the MIZ fraction of roughly 0.050-0.10 for March, 0.17-0.52 for July, 0.21-0.57 for August, and 0.4-0.15 for September. The correlations between the model and observations tend to be lower than the correlations between the observations themselves (Table 2). High correlations (>0.843) exist between the Bootstrap and AMSR relative MIZ extent values for all months examined, with generally lower values in July and August between Bootstrap and OSI-450.

## 4.3 Changes in MIZ location and geometry

There is no trend in the absolute MIZ extent (dashed lines in Fig. 1), but the location of the MIZ in the more recent years is further northwards, towards the pole (Fig. 3). The observational trends averaged over July, August, and September are consistent with those found in Strong and Rigor (2013) at 0.060, 0.056, and 0.059 degrees latitude per year for the Bootstrap, OSI-450, and Strong and Rigor (2013) datasets respectively. The model overestimates the latitude change at 0.117 degrees per year. There is close agreement in the average latitude change across the observations despite the fact that each timeseries cover slightly different temporal ranges, with the Strong and Rigor (2013) dataset covering the period from 1979-2011, and the other datasets covering from 1979 through 2017, 2015, and 2016 for the Bootstrap, OSI-450, and CICE-CPOM-2019 datasets, respectively. The individual trends (and RMS) in latitude for Bootstrap, OSI-450, and CICE-CPOM-2019, respectively (in degrees per year) are for July: 0.039 (0.387), 0.036 (0.484), 0.069 (0.806); August: 0.068 (0.607), 0.065 (0.667), 0.122 (0.998); September: 0.074 (0.708), 0.069 (0.896), 0.159 (1.13) degrees per year. The interannual variability of the mean latitude of the MIZ is roughly 10 to 30 times larger than the annual trends. The fractional changes in MIZ width required for the MIZ extent to remain constant have been calculated as described in Section 3.3 and show similarity to the fractional change in MIZ width as derived from sea ice concentration (Table 3). This is with the exception of the model which overestimates the MIZ width. In July, the required increase in the MIZ width for the approximated extent to remain constant is 10% for Bootstrap (over the period 1979-2017) and 9% for OSI-450 (1979-2015). This is compared to the fractional change in width of the MIZ based on average latitudes of the MIZ grid cells for Bootstrap and OSI-450 of 16% and an insignificant value respectively. In August, both the Bootstrap and OSI-450 datasets require a 20% increase in width to maintain MIZ extent as it moves northwards given our geometrical simplification, and have an average 24% and 25% increase in width from the observed average latitudes of their respective MIZ grid cells.

Although the MIZ is trending northwards, the observations do not support any trend in its overall sea ice area, with the exception of March for Bootstrap at -0.0025 x $10^6$ km$^2$ per year (Fig. 4). The modelled sea ice area within the MIZ did not show a trend except for July and September at 0.027 x $10^6$ km$^2$ per year and -0.0092 km$^2$ per year, respectively (Fig. 4). To

further illustrate the discrepancy of MIZ location between the observational datasets, we give the example of August 1993 (Fig. 5a) and August 2013 (Fig. 5b) spatial maps of MIZ contours. The spatial variability of the MIZ is poorly constrained by observations (Fig. 5). In 20 years the MIZ has shifted northwards, and the ice pack has become separated by stretches of MIZ. The similar ice extent contours (15% sea ice concentration, given by the solid lines in Fig. 5) illustrate that the similar magnitude of ice extent (Fig. 1) are also consistent with ice location. The pack ice contours (dashed lines in Fig. 5) show

differences between the datasets, accounting for the variability and differences in the MIZ extent (Fig. 1, Table 2). In 1993, the pack ice is not separated by areas of partial ice cover (Fig. 5a) as it is in 2013 (Fig. 5b). The MIZ covers more of the central Arctic in the more recent year (2013) than it does in 1993. The lack of trend in MIZ extent is robust given changes in the upper and lower bounds of the sea ice concentration thresholds in the MIZ definition.

## 5 Discussion

**5.1 Differing definitions of MIZ extent**

    Similar to sea ice extent, the MIZ extent is also defined by sea ice concentration thresholds. Another definition of the MIZ in common usage is that the MIZ (e.g. Squire, 2020) is that region of partially-ice covered ocean that is impacted by ocean waves. One drawback of this definition is that it necessitates further definition of where the ice-covered ocean is deemed to be 'impacted by ocean waves'. This could be problematic because different applications (e.g. shipping, climate studies) could

require different thresholds of when they consider waves important. There are also significant uncertainties with both observing and forecasting waves within the sea ice and this is an ongoing field of study (Roach et al., 2019; Stopa et al., 2018). For instance, it has been shown that ocean waves can penetrate deeper into the ice pack than previously thought (Kohout et al., 2014). Although the definition of the MIZ using ocean wave penetration can be very useful for other studies (for example, boundary layer air-sea interaction or wave-action studies), we argue that comparisons of purely MIZ extent from different

observational datasets and models should be done through sea ice concentration thresholds. This is especially true for model comparisons given the unknowns in wave-sea ice interaction (Squire, 2020). Some techniques used to analyse total sea ice extent such as geographical muting (Eisenman, 2010) only apply to those months where sea ice extends beyond the limit of the land, if the land was not present. During the summer months, the geographical muting would not well explain why the MIZ extent remains constant.

**5.2 Trends and correlations between observations and model**

    The lack of trend in the MIZ extent contrasts with the significant decline in total sea ice extent (Fig. 1, Table 1). While September is a common month to examine for projecting future sea ice extent since it is the month of the year where sea ice reaches its annual minimum (Comiso et al., 2008), it is interesting to note that for studies of the MIZ, it is July and August which may be more informative because these months show the greatest differences in trends of MIZ fraction between the

observational and model results (Table 1). These seasonal differences in observations of the MIZ fraction and model result

will have consequences for any future projections of the MIZ, and one must be wary of monthly extrapolation in particular during the summer months.

The size of the MIZ is poorly defined by observations, and it follows that models of the MIZ can only be constrained within these observational values. There have been recent developments in modelling of the MIZ, such as how waves break up the ice (Kohout et al., 2014; Montiel and Squire, 2017), the simulation of the floe size distribution and changes of sea ice floe size (Roach et al., 2018), and how sea ice floe size information is important for accurately capturing the seasonality of sea ice concentration in climate models (Bateson et al., 2020). However, the results in this study highlight the fact that attention must also be given to improving observations of the MIZ location and extent in order to validate such models. It is important to note that while the relative MIZ extent is slightly increasing due to decreasing total sea ice extent, it does not necessarily follow that the MIZ extent itself is also increasing. The lack of trend in the MIZ extent gives an indication about how the sea ice is melting.

Given that the sea ice area is declining, it could be (and is often assumed) that the sea ice concentration is declining everywhere. However, we have found no trend in the observations of sea ice area in the MIZ except for the slight negative trend in March in the Bootstrap data, but the spread of the sea ice area within the MIZ across the observational datasets is large (Fig. 4). Due to this, there could possibly be a trend in the MIZ sea ice area which we are not able to resolve. For example, the slight significant trends of sea ice area in the MIZ shown by the model are still within the range of observations. Since there is no trend in sea ice area within the MIZ and no trend in the MIZ extent, there is no significant change of sea ice concentration within the MIZ based on observations (where sea ice concentration in the MIZ is given as the ratio of the area of sea ice in the MIZ and the extent of the MIZ). Similarly, there would not be any trend of sea ice area within the MIZ relative to the MIZ extent. Since there is also no observed change in MIZ extent, it follows that the pan-Arctic averaged sea ice concentration is not declining in concert with its declining extent. This suggests that changes to the extent of the MIZ depend strongly on the sea ice thickness distribution.

## 5.3 MIZ trending northward

Since the MIZ extent remains constant, it then follows that the central pack ice extent is decreasing because the total ice extent is decreasing (Fig. 1). More specifically, the inner pack ice area is outpacing the decline of total ice area, causing a widening trend (Strong et al., 2017). Because the width of the MIZ is increasing in summer (Strong and Rigor, 2013) while the total extent remains constant then the perimeter around the MIZ must be decreasing, forcing a northward movement (Fig. 3, 4). This is consistent with the positive trend in the area-weighted latitude of the MIZ found for the same months with the same MIZ definition in Strong and Rigor (2013). This northward migration of the MIZ has broad implications for changes in the coupled bio-geo-physical climate system.

A declining sea ice cover in summer is a main contributor to the amplification of increasing temperatures in the Arctic (Screen and Simmonds, 2010). The MIZ is also a potential area for Arctic cyclogenesis, which allows for significant heat release from the ocean to the atmosphere (Inoue and Hori, 2011), thus contributing to the temperature amplification. With a northward shifting storm track (Sepp and Jaagus, 2011), a northward shift of meridional heat transport is also expected. In addition, changes in MIZ location will have regional implications for total momentum transfer from the atmosphere to the ocean through the ice, because maximum momentum transfer occurs at moderate ice concentrations (~70-90%), full ice cover, and low ice concentrations (~10-30%) (Cole et al., 2017; Tsamados et al., 2014) and is also impacted with varying surface roughness (Martin et al., 2016).

From a biological perspective, it has already been established that sea ice receding further from the coastline, followed by the MIZ (Fig. 3), is a problem for marine mammals who use the sea ice as a platform for resting, hunting and breeding (Hamilton et al., 2015; Kovacs et al., 2011). When there is no ice to rest on, there have been increasing accounts of animals changing their behaviour to use land as a refuge. For example, walrus have been increasingly observed in mass haul-outs (Jay et al., 2012) resulting in premature death due to overcrowding. Other important impacts of the northward-trending MIZ on sea ice-associated biota have been explored. For instance, the northward movement of the MIZ has an impact on primary productivity of sea ice algae due to changes in light availability (Tedesco et al., 2019). Ice algae grow on the underside of (and within) the sea ice and are an early important food source for zooplankton and ice fauna (Hegseth, 1998; Horner et al., 1992; Søreide et al., 2013). However, one aspect that could be further explored is the impact of an unchanging MIZ extent in combination with the northward movement of the MIZ. The extent provides a metric about the range of the habitat for MIZ-dependent animals. For example, the deformed ice in the MIZ creates ridged habitats underwater for animals such as polar cod (Hop and Gjøsæter, 2013), and also habitats above the ice for animals such as seals, polar bears, and seabirds (Hamilton et al., 2017).

**5.4 Increase in width compensates for decrease in perimeter**

Given our simplifications for MIZ geometry, the fractions of the required changes in MIZ width in order for the MIZ extent to remain unchanged (first row of Table 3) are relatively consistent with the calculated fraction change of MIZ width from sea ice concentration data (second row of Table 3), with the exception of the model. The model is showing a greater increase in MIZ width than the observations, with the greatest overestimation of MIZ width occurring in July. This monthly variation in how much the model overestimates MIZ width could lead to other overestimations that would then also vary by month, such as an overestimated atmosphere-ocean heat transfer in July. The similarities of the observed fraction change in MIZ width and necessary fraction change for a constant MIZ extent from the observational datasets (Table 3) provide support that the MIZ is widening enough to maintain its extent as it travels northwards and its perimeter decreases.

### 5.5 Sources of error

Observational uncertainty is one factor among others which must be considered when assessing the accuracy of any model (Notz, 2015), including the CICE-CPOM-2019 model used here. An error of 10% applied to the observational products in this study is consistent with other studies, as noted in Section 3.2. The error of 10% has been chosen because it is consistent with the systematic differences between the ASI algorithm used to generate the AMSR data and other observational products (NASA Team 2 and Bootstrap) are approximately 10% (Spreen et al., 2008). It is clear from the differences in the observations

that the uncertainty varies seasonally and often exceeds 10%, with the greatest uncertainty in August (Fig. 2 and 3). Comiso and Steffen, (2001) found an error range between visible/infrared-derived ice concentrations (e.g. AVHRR) of 5-20%. The error between AVHRR products and other SSM/I products ranging from 0.7 and 10.5% (with 5.3% error between AVHRR and Bootstrap) (Meier, 2005). A source of error for SSM/I concentrations is the use of hemispheric tie-points, which are unchanging and may not agree on conditions at a specific time and place (Meier, 2005). As well, since SSM/I concentrations

are calculated based on daily composites of brightness temperatures and then averaged onto a 25-km resolution grid, it will result in errors stemming from spatial and temporal averaging (Meier, 2005). Our study reveals that the systematic error in deriving the MIZ from these satellite products must be larger than 10% as documented by differences in monthly mean MIZ values of up to 300% (Fig. 1).

### 6 Conclusions

We have analyzed the evolution of the absolute and relative marginal ice zone from 1979 through 2017 based on four satellite retrievals (OSI-450, Bootstrap, AMSR-E, and AMSR-2) and simulations with a stand-alone sea ice model CICE-CPOM-2019 including a floe size distribution model. While all products agree within their uncertainties during winter, large discrepancies occur during summer between the satellite products. We have found no significant trend in the MIZ extent across any of the observational datasets examined here (OSI-450, Bootstrap, and AMSR), with the exception of a small negative trend in March

for Bootstrap. Due to the decrease in Arctic sea ice extent, there is a significant increase ( > 50% ) in the relative MIZ extent (MIZ extent divided by sea ice extent) during August and September for the Bootstrap and OSI-450 observational datasets. During July and August, the positive trend is 2 to 4 times stronger in our model simulation than these observations. We found no observed trend in the sea ice area within the MIZ (except for a slight negative trend in March for the Bootstrap dataset), but the observed spread of sea ice area within the MIZ is too great such that the significant trend of the modelled sea ice area in

the MIZ still lies within the spread of observations. Due to the also large spread in the observations in MIZ extent, we should be cautious about what conclusions we make about whether or not there is a true trend in the MIZ extent, and how well we can validate our MIZ models. Given this uncertainty, the fact that climate model projections show the Arctic becoming seasonally ice free by the mid-century (Notz and Stroeve, 2018) does not mean we will have an increased area of the ocean covered by marginal ice as defined by the 15-80% ice cover threshold definition. Only at the point when there is a completely seasonal

ice cover in conjunction with no pack ice, would our results suggest that further ice loss will result in decreases in MIZ extent.

*Author contribution:* RR performed the model simulations and wrote the code for analysis. RR wrote the manuscript with contributions from DF and DS.

*Acknowledgements:* This research was funded by the National Environment Research Council of the UK (NERC), Award number NE/R000654/1. We would like thank Courtenay Strong, Chris Horvat, and 2 other anonymous reviewers for their useful comments that helped improve the manuscript. We would also like to thank the editor, John Yackel.

*Competing interests:* Daniel Feltham and David Schröder are members of the editorial board of the journal.

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

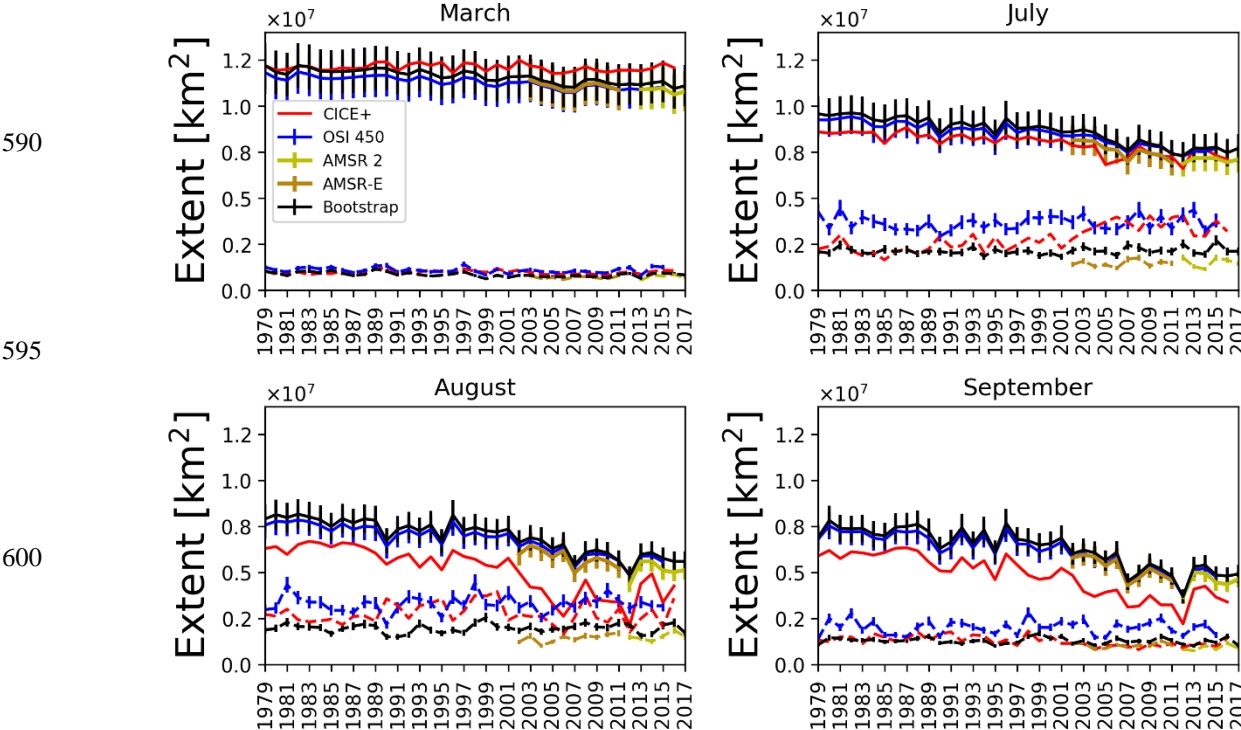

**Figure 1:** **Arctic sea ice extent (solid lines) and marginal ice zone extent (dashed lines) from our CICE simulation CICE-CPOM-2019 and four remote sensing products for (a) March (b) July (c) August (d) September. Marginal ice zone extent is defined as the area where sea ice concentration is between 15-80%. Sea ice extent is the area of ice coverage above 15%. An error bar of 10% has been applied to the observational output.**

610

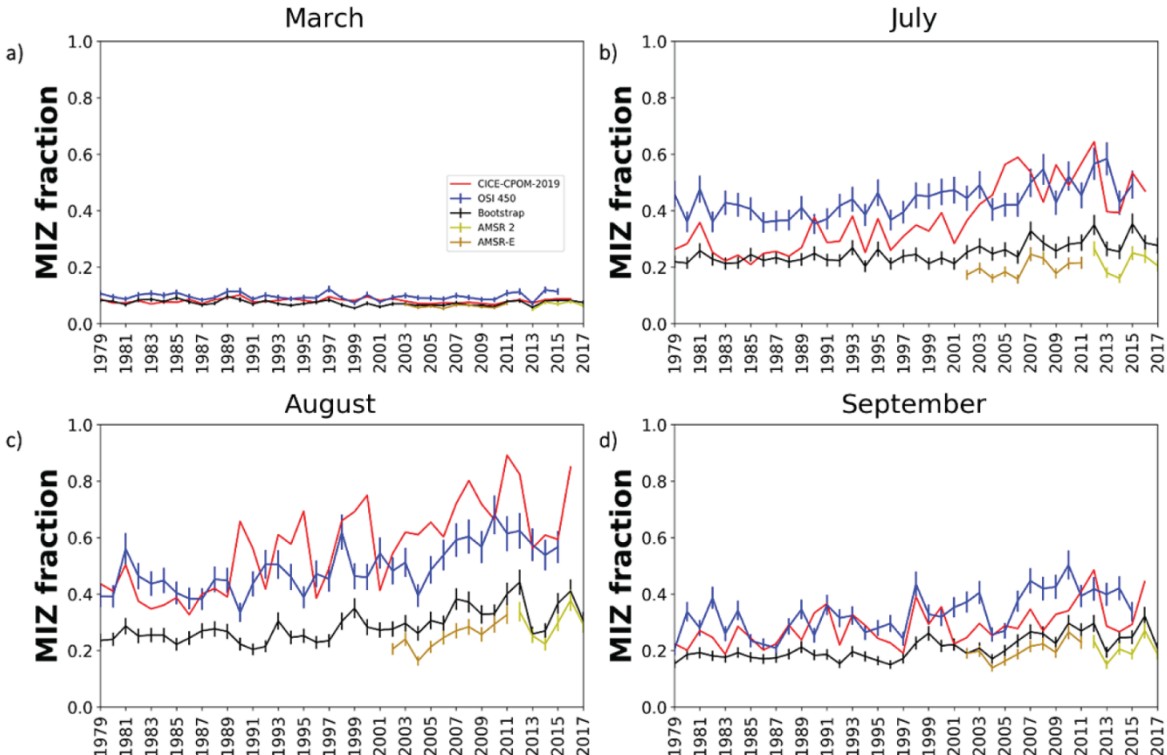

**Figure 2: As Figure 1, but MIZ fraction for (a) March (b) July (c) August (d) September. Marginal ice zone extent is defined at the area where sea ice concentration is between 15-80%. Sea ice extent is the area of ice coverage above 15%. An error bar of 10% has been applied to the observational output.**

615

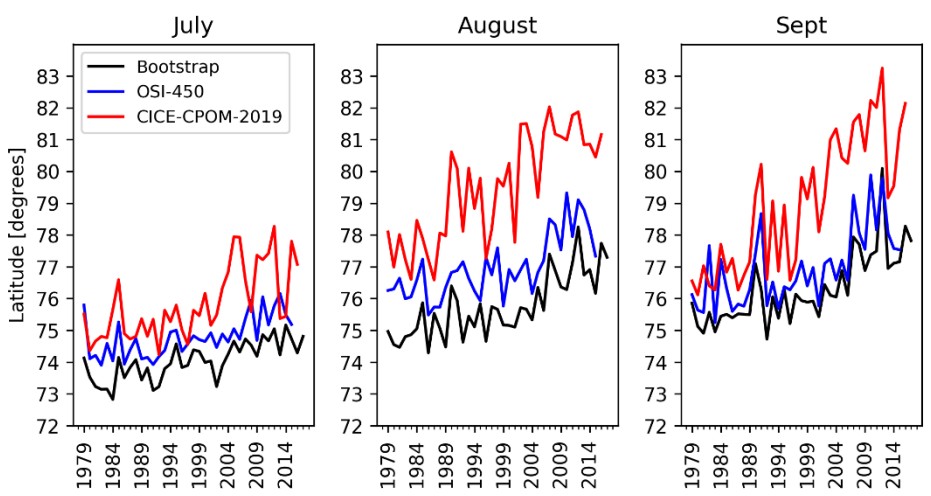

**Figure 3: Timeseries of monthly-averaged latitudes of MIZ for Bootstrap (black), OSI-450 (blue) and model CICE-CPOM-2019 (red).**

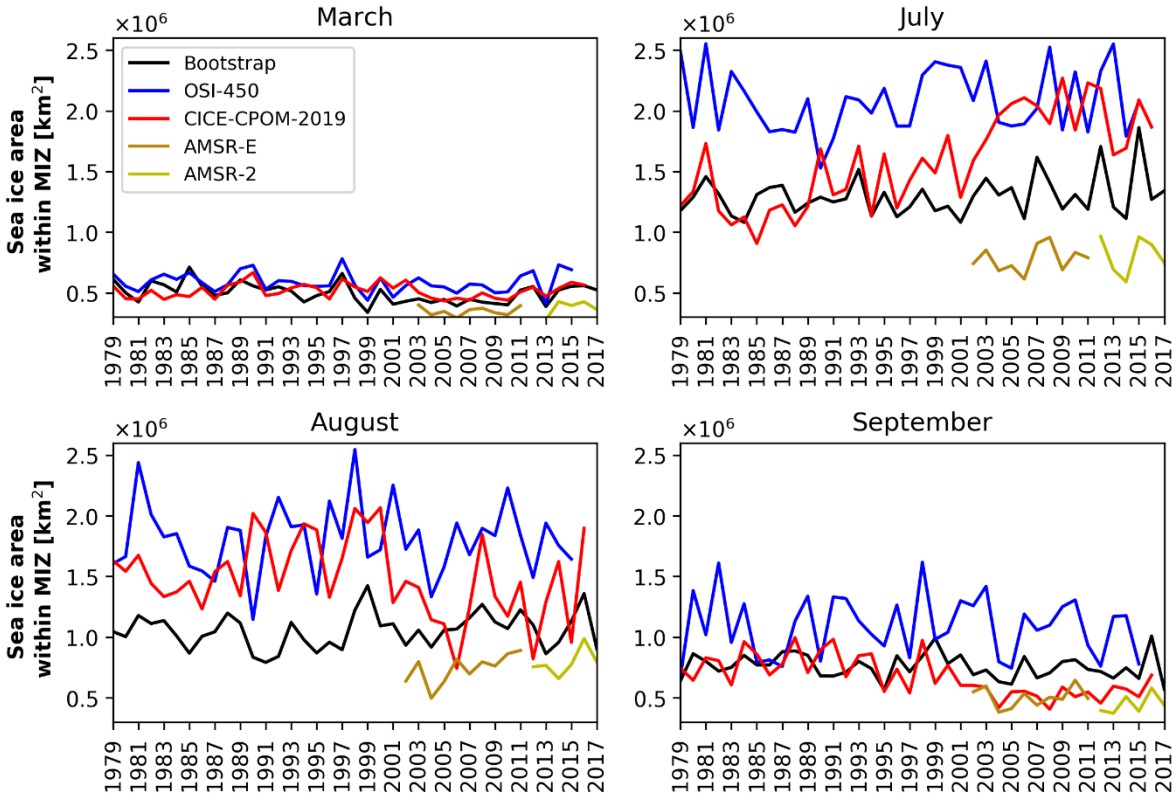

**Figure 4: Timeseries of sea ice area within the MIZ, monthly-averages from daily data.**

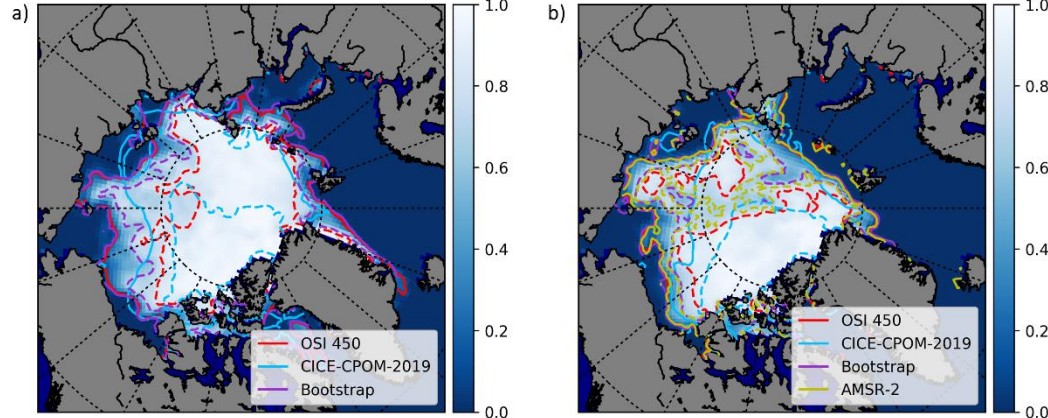

**Figure 5: The location of the MIZ for August 1993 (a) and August 2013 (b). The MIZ is trending northwards in recent years (See also Figure 3 for timeseries of latitudes). Dashed and solid lines represent the 80% and 15% sea ice concentration levels, respectively. AMSR data is not available prior to 2002. Underlying sea ice concentration is from the OSI-450 satellite product.**

α = 0.05

| Trend in $10^{10}$ m$^2$ per year (r$^2$) | | | Relative MIZ | Total change |
|---|---|---|---|---|
| **March** | Total ice extent | MIZ extent | (MIZ extent/total ice extent) [1/year] | Relative MIZ (%) |
| **OSI 450** | -2.42 (0.74) | Insig. | Insig. | Insig.* |
| **Bootstrap** | -2.76 (0.78) | -0.520 | Insig. | Insig.* |
| **AMSR** | -3.04 (0.43)* | Insig.* | Insig.* | Insig.* |
| **CICE-CPOM-2019** | Insig. | Insig. | Insig. | Insig.* |
| **July** | | | | |
| **OSI 450** | -5.27 (0.84) | Insig. | +0.003 (0.375) | 27% |
| **Bootstrap** | -5.85 (0.87) | Insig. | +0.002 (0.450) | 38% |
| **AMSR** | -7.55 (0.67)* | Insig.* | Insig.* | Insig.* |
| **CICE-CPOM-2019** | -4.29 (0.70) | Insig. | +0.009 (0.636) | 124% |
| **August** | | | | |
| **OSI 450** | -6.52 (0.78) | Insig. | +0.005 (0.479) | 50% |
| **Bootstrap** | -7.19 (0.81) | Insig. | +0.003 (0.444) | 56% |
| **AMSR** | -7.96 (0.47)* | Insig.* | +0.008 (0.672)* | 60% |
| **CICE-CPOM-2019** | -9.61 (0.71) | Insig. | +0.010 (0.557) | 91% |
| **September** | | | | |
| **OSI 450** | -7.80 (0.75) | Insig. | +0.004 (0.392) | 79% |
| **Bootstrap** | -8.07 (0.75) | Insig. | +0.003 (0.479) | 66% |
| **AMSR** | -9.72 (0.50)* | Insig.* | Insig.* | Insig.* |
| **CICE-CPOM-2019** | -9.02 (0.79) | -1.37 (0.31) | +0.003 (0.293) | 57% |

Table 1: Trends with r$^2$ values in brackets for total sea ice extent, MIZ extent, and extent of the MIZ relative to the total sea ice extent (also as a total % change) for the model run and all observational datasets examined. Note that the periods between above datasets are not the same: OSI-450 (1979-2015), CICE-CPOM-2019 (1979-2017), Bootstrap (1979-2017), and AMSR (June 2002 – 4 Oct 2011 AMSR-E, July 2012 – 2017 AMSR2). The AMSR trends are denoted with * to clearly indicate the shortened time coverage of those observations in comparison with the rest.

α = 0.05

| Correlations | Total ice extent | | | MIZ extent | | | Relative MIZ extent | | |
|---|---|---|---|---|---|---|---|---|---|
| **March** | **OSI 450** | **CICE19** | **Bootstrap** | **OSI 450** | **CICE19** | **Bootstrap** | **OSI 450** | **CICE19** | **Bootstrap** |
| **CICE-CPOM-2019** | 0.448 | | | 0.548 | | | 0.502 | | |
| **Bootstrap** | 0.998 | 0.440 | | 0.894 | 0.415 | | 0.876 | 0.392 | |
| **AMSR\*** | 0.957 | Insig. | 0.962 | 0.922 | 0.759 | 0.889 | 0.932 | 0.792 | 0.902 |
| **July** | **OSI 450** | **CICE19** | **Bootstrap** | **OSI 450** | **CICE19** | **Bootstrap** | **OSI 450** | **CICE19** | **Bootstrap** |
| **CICE-CPOM-2019** | 0.896 | | | Insig. | | | 0.547 | | |
| **Bootstrap** | 0.996 | 0.913 | | 0.362 | 0.414 | | 0.655 | 0.778 | |
| **AMSR\*** | 0.989 | 0.575 | 0.990 | 0.595 | Insig. | 0.869 | 0.645 | Insig. | 0.922 |
| **August** | **OSI 450** | **CICE19** | **Bootstrap** | **OSI 450** | **CICE19** | **Bootstrap** | **OSI 450** | **CICE19** | **Bootstrap** |
| **CICE-CPOM-2019** | 0.902 | | | Insig. | | | 0.539 | | |
| **Bootstrap** | 0.998 | 0.900 | | 0.472 | Insig. | | 0.719 | 0.749 | |
| **AMSR\*** | 0.991 | Insig. | 0.987 | 0.826 | Insig. | 0.637 | 0.886 | 0.755 | 0.843 |
| **September** | **OSI 450** | **CICE19** | **Bootstrap** | **OSI 450** | **CICE19** | **Bootstrap** | **OSI 450** | **CICE19** | **Bootstrap** |
| **CICE-CPOM-2019** | 0.957 | | | Insig. | | | 0.552 | | |
| **Bootstrap** | 0.998 | 0.955 | | 0.577 | Insig. | | 0.755 | 0.719 | |
| **AMSR\*** | 0.990 | 0.778 | 0.989 | 0.805 | Insig. | 0.852 | 0.780 | 0.675 | 0.927 |

Table 2: Correlations of the inter-annual variability for the total sea ice extent, MIZ extent, and extent of the MIZ relative to the total sea ice extent for the model run and all observational datasets examined. The AMSR trends are denoted with * to clearly indicate the shortened time coverage of those observations in comparison with the rest.

| | July | | | August | | | September | | |
|---|---|---|---|---|---|---|---|---|---|
| Required fraction change of MIZ width for MIZ area to remain constant | **Bootstrap** | **OSI-450** | **CICE-CPOM-2019** | 1.20 | 1.20 | 1.56 | 1.23 | 1.21 | 1.75 |
| | 1.10 | 1.09 | 1.20 | | | | | | |
| Calculated fraction change from MIZ width trends | 1.16 | Insig. | 2.42 | 1.24 | 1.25 | 1.49 | 1.17 | Insig. | Insig. |

**Table 3: Fraction changes of MIZ width needed for the MIZ area to remain constant compared with the calculated trends in MIZ width assuming an averaged perimeter**