# Peer review of "Changes of the Arctic marginal ice zone during the satellite era"

_The Cryosphere, 2019_

## Referee Comment (RC1) · Anonymous Referee #1 · 6 Dec 2019

This paper shows that there is no trend in the areal extent of the marginal ice zone (MIZ), an increase in the fractional area that the MIZ covers in the total sea ice extent, and that the CICE-CPOM model fails to reproduce these observations.

I think the observation that the total areal extent of the MIZ hasn't changed is an interesting way to reconsider the dramatic changes in Arctic sea ice, but this isn't really a new insight. For example, Strong and Rigor (2013) and other studies have shown that the MIZ moved northward and its width has increased. Rolph et al. is simply arguing that the glass is half full (no change in MIZ extent), rather than half empty (MIZ width is increasing). While I think this is an interesting way to look at the changes in Arctic sea ice, does this different perspective provide any new scientific advances? The authors also need to consider that the sea ice concentration data has larger errors during summer than they assume. As this paper currently stands, I don't think it provides enough compelling reason to warrant publication.

[Figure]

Major Comments:

1) Why is it important to consider that the areal extent of the MIZ hasn't changed? The authors need to beef up their case that it is important to think of the changes in the MIZ this way. Can the authors show how this perspective provides new insights that the many physical process studies of changes in the fractional area of young ice versus old ice do not? Or new insight into some biological process?

2) The errors in the sea ice concentration retrievals from passive microwave satellites during summer are large. For example, in their figure 3 they show wildly varying estimates of where the northern edge of the MIZ is. Some (Walt Meier and/or others at NSIDC or NASA may have a paper on this) have estimated the summer SIC error to be higher than 40% during summer, and most of this error and differences between the retrieval methods is related to how they filter weather. Rolph et al. need to provide a more thorough error analysis than assuming an overall 10% error estimate since the errors in the SIC retrievals affect how robust their conclusions are.

3) The fact that models don't reproduce these observations isn't surprising. There are already many papers that show that various models don't reproduce some observation. But as with any tool, does simply showing that a tool doesn't work for this job warrant publication? If Rolph et al. could pin down what needs to be improved in the models, that would advance science and the inclusion of the model study would be interesting.

Minor Suggestions:

4) Be consistent in your use of units. E.g. in lines 194-195 you switch between meters squared to kilometers squared. I suggest sticking with kilometers squared.

5) Need to note $10^7$ in the label for the Y axes in Fig. 1 rather than "1e7" on the top corner of the plots.

6) Provide a short section 3.3 discussing how statistical significance was estimated. Maybe just move this from caption of table 1.

7) Caption of Fig. 1: Change ". . .is defined at. . ." to ". . .is defined as. . .".

---

## Referee Comment (RC2) · Court Strong (Referee) · 11 Dec 2019

The authors present an analysis of historical MIZ extent using available satellite products and the CICE-CPOM model. They find no historical trend in extent but an increase in the fraction of the total ice that is MIZ. MIZ extent provides an interesting perspective which is complementary to the previously published trends in MIZ position and width. Within the scope of the present study, an explanation for the lack of trend drawing on MIZ geometry and prior results could strengthen and contextualize the findings.

Major comments:

1. A poleward trending and widening MIZ does not necessarily need to conserve area, so the lack of trend reported here is potentially interesting. The manuscript

would be strengthened by explaining how this result follows from the magnitude and direction of changes in MIZ width and position. One could, for example, simplify the geometry by approximating the MIZ as an annulus and then plug in the latitude rate of change (as a radius) and width rate of change from Table 1 of Strong and Rigor (2013). Over the satellite record, this gives changes in warm-season MIZ extent which are small relative to interannual variability.

2. Related to above, the authors touch on the concept of perimeter briefly in their remarks on lines 41 and 260, but this can be made more quantitative and also contextualized by prior related work. For example, Strong et al. (2017) calculated pan-Arctic MIZ extent in the bootstrap data, denoted by $A_\Omega$ in their equation (15), and used this time series in conjunction with MIZ perimeter ($\overline{L}$) to study the width trend. They also concluded that the widening is consistent with the decline in the inner pack ice area outpacing the decline in total ice area (expressed as effective radii; trends reported at the end of their Section 4a and Fig 8b).

3. Section 3.1: For model validation, the interpolation of concentration onto the model grid makes sense. However, to provide a definitive statement on MIZ extent trends, why not use the native ∼25-km NSIDC grid? I think the nominal resolution around the pole in the 1-degree tripolar grid is about 85 km, although line 100 in Section 2.2. mentions ∼40 km. Either way, potential artifacts of the re-gridding and interpolation should be considered because MIZ width ranges from about 50 to 150 km.

4. The abstract states that the MIZ is "trending northwards" and Section 4.3 is titled "MIZ trending northwards," but the presented results seem restricted to maps of August 1993 and August 2013. I did not see the record-length analysis to support the statement in the abstract "The MIZ is trending northwards, consistent with other studies" (line14).

5. The MIZ fraction change is reported as "small" in the abstract, and a quantitative

value would be informative here. Also, is it really small? If I understand the units correctly, a 0.003 / year trend would amount to an increase of 0.117 MIZ fraction over the record. For a quantity starting around 0.2, increasing to 0.3 would be a 50% increase.

6. We see that the model performance varies through the year as discussed in Section 4.1, but it is difficult to interpret the discrepancy from the warm-season observations because the spatial pattern is left implicit. Does the total extent error signal that the model MIZ has a position error, width error, or both? A more spatially explicit treatment of the model performance would help the reader to understand the purpose of including the model, and its intended role and weight in the suite of results.

7. Suggest including a paragraph somewhere in main text to detail the statistical methods (assumed degrees of freedom, tests were parametric versus bootstrap, etc.).

8. The title is very general. To more precisely reflect the presented analysis, suggest something like: "Historical analysis of Arctic marginal ice zone extent".

Minor comments:

1. Line 11 in abstract: I did not see an extrapolation of the results forward in time in the paper. If this remark just follows from the report of no trend, suggest removing to avoid implying that a supporting extrapolation with uncertainty analysis was performed.

2. Lines 14-16 recommends that future authors "provide a specific and clear definition when stating that the MIZ is rapidly changing." Suggest an edit here to clarify if future authors are being asked to specify the MIZ definition or to specify the particular MIZ property that is changing (width, area, latitude, etc.).

[Figure]

3. Lines 22-24 state that the cited studies "tend to assume that marginal ice zone (MIZ) extent is increasing." I am familiar with these studies and looking back through a few of them as a sample, I found no assumption that MIZ extent is increasing. Instead, the remarks about MIZ change were literature-based and referred to specific properties.

4. Why was the NSIDC Climate Data Record not used? I think one of the motivations for CDR was to develop a consistent record suitable for trend analysis.

5. Line 202: It's not clear what is meant by "The interannual variability of the MIZ ... varies more than the sea ice extent." A more precise statement referencing specific variance statistics could clarify.

6. Line 212 and thereafter. Suggest using a consistent format when referring to the MIZ fraction trends. Something like "0.003 per year" as in the Table seems less likely to confuse than 0.3% (the latter could be interpreted as a percent change rather than change in percent).

7. Line 238: "Our results are robust" – not clear which specific results are referred to here.

---

## Referee Comment (RC3) · Anonymous Referee #3 · 19 Dec 2019

General comments

The manuscript "Changes of the Arctic marginal ice zone" by R. Rolph, D. Feltham, and D. Schröder provides a clear analysis of evolution in Arctic marginal ice zone (MIZ) extent relative to total sea ice extent (SIE) in a changing climate. In highlighting, based on an operational definition, that the MIZ extent shows no significant trend over the last 40 years despite a decline and well-defined trend in total SIE, this analysis underscores the need for a universal definition for the MIZ, identification of relevant variables in addition to extent for its characterization, and improved understanding of implications in a changing climate for communities influenced by MIZ processes.

This paper addresses relevant scientific questions including characterization of the MIZ, and presents novel analysis that contributes to an understanding of changes in the sea ice cover, and in particular poleward migration in MIZ and total SIE, in the

context of a changing climate. Also of interest however is the sensitivity of this analysis to the mathematical and physical definition for the MIZ; investigation of additional techniques used to analyse total SIE (i.e. geographic muting described in Eisenman, 2010) applied to the MIZ that could perhaps explain the absence of statistically significant trends in MIZ extent over the past 40 years and, as noted by other reviewers; further exploration of reasons for the absence of changes in MIZ extent; in addition to alternative MIZ variables/aspects (area, regional variability, zonal mean MIZ edge as in Eisenman, 2010) that do reflect changes in the zone between fully ice-covered and ice-free regions in response to global warming. This is therefore to recommend that the manuscript be published following revisions that address MIZ definitions and analysis. Please find below more specific comments for consideration.

Specific comments Abstract

p. 1, lines 6 – 8. "It does not logically follow, however, that the extent of the marginal ice zone (MIZ), here defined as the area of the ocean with ice concentrations from 15 to 80%, is also changing". What are the implications of assumptions associated with a changing MIZ extent?

p.1, lines 14-16. "Given the results of this study, we suggest that future studies need to remain cautious and provide a specific and clear definition when stating the MIZ is 'rapidly changing'." Perhaps provide an appropriate definition and context for the statement of a 'rapidly changing' MIZ. As is noted below, additional MIZ definitions and changes in additional MIZ characteristics over the past 40 years could be evaluated and compared with MIZ extent to determine whether these properties and attributes capture a rapidly changing MIZ.

Introduction

p. 2, line 45. Perhaps include 'extent' following 'MIZ'.

p. 2, lines 45 – 46. "It also follows that we need to be aware of the extent to which our

observations are able to constrain any model of the MIZ". Does this study also highlight the need for a universal and/or alternative definition for the MIZ?

p. 2, line 57. "Here we also describe how we defined the MIZ and sea ice cover in our calculations". Will the results from this analysis differ for different MIZ definitions?

p. 2, line 58. The timeframe could be indicated following "March, July, August, and September".

Methods

p. 6, lines 167 – 170. Perhaps the MIZ area could be examined in addition to MIZ extent, and results compared to characterize changes relative to total SIE and area over the past 40 years.

p. 6, lines 176 – 177. "…an error of 10%…" Does this uncertainty vary seasonally?

p. 6, lines 177 – 178. Perhaps conduct the same analysis for sea ice area, MIZ area, and relative MIZ area.

Results

p. 7, line 195, and p. 8, line 230. Absence of trend in MIZ sea ice extent and northward migration in MIZ. The absence of statistically significant trends in MIZ extent suggests poleward migration of the southern and northernmost MIZ boundaries at compara-ble rates. Application of the zonal-mean sea ice edge concept outlined in Eisenman (2010) to the northernmost and southernmost boundaries (in a sense converse to the SIE analysis, since with a deteriorated sea ice cover the northern boundary is less sta-ble and muting less pronounced) would illustrate rates of change for each, as well as regional variability. Also of interest is the transition to lower sea ice concentrations in the MIZ over the past 40 years, documented by MIZ area. Please see also comments pertaining to the Discussion.

Discussion

p. 9, line 256. Perhaps include the phrase 'due to decreasing total SIE' following "slightly decreasing".

p. 9, line 262. Northward migration in the poleward MIZ boundary and area-weighted latitude of the MIZ. Also of interest is the study by Eisenman (2010) describing the role of zonal mean ice edge latitudes in describing asymmetry in winter and summer decline in SIE, in addition to the study by Stroeve et al. (2016) implementing a similar concept to define Antarctic MIZ boundaries according to zonal mean latitudes based also on the approach outlined in Strong and Rigor (2013). It would be interesting to see how evolution in the i) northern and ii) southern latitude MIZ boundaries/edges and iii) area (rather than extent, based on discussions outlined in Notz; 2014) bounded by each, compares with results from the present analysis based on MIZ extent, and whether this approach captures asymmetry in the seasonal cycle as well as rates of poleward migration in the northern and southern MIZ boundaries. Evaluation of MIZ area might also illustrate the nature of transition to a lower sea ice concentration regime in the MIZ over the past 40 years.

Conclusions

p. 10, lines 300-303. "Due to the spread of the observations in MIZ extent..." As previously noted, context for the phrase 'rapidly changing' should be provided (i.e. extent and/or other MIZ aspects including northern and southern MIZ boundaries and area).

Technical corrections

p. 8, line 237. Please remove 'is'.

p. 10, line 295. Perhaps replace 'big' with 'large'.

References

Eisenman, I., 2010: Geographic muting of changes in the Arctic sea ice cover, Geophys. Res. Lett., 37, L16501, doi:10/1029/2010GL043741.

Notz, D., 2014: Sea-ice extent and its trend provide limited metrics of model performance, The Cryosphere, 8, 229–243, https://doi.org/10.5194/tc-8-229-2014.

Stroeve, J. C., Jenouvrier, S., Campbell, G. G., Barbraud, C., and Delord, K., 2016: Mapping and assessing variability in the Antarctic marginal ice zone, pack ice and coastal polynyas in two sea ice algorithms with implications on breeding success of snow petrels, The Cryosphere, 10, 1823–1843, https://doi.org/10.5194/tc-10-1823-2016.

Thank you for the opportunity to review this manuscript.

Please also note the supplement to this comment:
https://www.the-cryosphere-discuss.net/tc-2019-224/tc-2019-224-RC3-supplement.pdf

---

## Author Comment (AC1) · 26 Mar 2020

**Reviewer 1.**

We thank the reviewer for their comments and provide our response below in blue.

This paper shows that there is no trend in the areal extent of the marginal ice zone (MIZ), an increase in the fractional area that the MIZ covers in the total sea ice extent, and that the CICE-CPOM model fails to reproduce these observations. I think the observation that the total areal extent of the MIZ hasn't changed is an interesting way to reconsider the dramatic changes in Arctic sea ice, but this isn't really a new insight. For example, Strong and Rigor (2013) and other studies have shown that the MIZ moved northward and its width has increased.

It is true that Strong and Rigor (2013) showed that the width of the MIZ was increasing and moving north. However, our manuscript shows, for the first time and within the bounds of observation error, that there is no trend in the MIZ extent. This is a new insight, as noted by the other reviewers, including the first author of the Strong and Rigor paper.

Rolph et al. is simply arguing that the glass is half full (no change in MIZ extent), rather than half empty (MIZ width is increasing). While I think this is an interesting way to look at the changes in Arctic sea ice, does this different perspective provide any new scientific advances?

Our analysis indicates that the MIZ extent is both not changing in extent *and* is increasing in width, in contrast to the reviewer's description of our results. It is not our intention to speak in sweeping terms about a 'glass half-full' or 'half-empty' situation in relation to climate change and sea ice, but rather to present the first historical analysis of the marginal ice zone extent, which is a vital part of the Arctic climate and biology, e.g. Barber et al. (2015).

The authors also need to consider that the sea ice concentration data has larger errors during summer than they assume. As this paper currently stands, I don't think it provides enough compelling reason to warrant publication.

Naturally, the error in summer sea ice concentration is larger than the 10% error bar we applied. This is evident from the fact that the different observation products do not agree within 10%. We have now made this point more explicitly (see below). However, the true error associated with each observation product is not a known quantity. There are complexities in the processing chain for each observation product produced and, while errors may be quoted for each step in the analysis chain, the true error in representation of sea ice concentration may be subject to systematic or random errors that are not fully accounted for. It is for this reason that we followed precedent and used the 10% error previously introduced in Spreen et al (2008).  Increasing the uncertainty of the sea ice concentration datasets would not lead to a known trend, given that the lower uncertainty we used does not show significant trends in MIZ extent.

Major Comments:
1) Why is it important to consider that the areal extent of the MIZ hasn't changed? The authors need to beef up their case that it is important to think of the changes in the MIZ this way. Can the authors show how this perspective provides new insights that the many physical process studies of changes in the fractional area of young ice versus old ice do not? Or new insight into some biological process?

- The Arctic sea ice area is declining with the strongest rate during summer. This can be described by either of the following two extreme scenarios: 1) sea ice concentration is reducing everywhere, so the whole Arctic will become MIZ before it will be free of ice, or 2) the sea ice concentration remains between 80-100% (our definition of pack ice), but the total sea ice extent is reducing until all of the ice is gone. There is no MIZ in the second scenario. We have shown that reality is somewhere in the middle. This is important to know because both of these extreme scenarios are physically very different. In the second scenario, sea ice thickness is homogenous within a grid cell, but in the first scenario, there is a wide sub-grid cell ice thickness distribution, with the thinner ice melting and thicker ice surviving. The changes to the extent of the MIZ depend strongly on the sea ice thickness distribution and provide insights to how sea ice can be expected to melt in the future. We have added a statement to the Discussion section 5.2, at line 324: 'The lack of trend in the MIZ extent gives an indication about how the sea ice is melting. Given that the sea ice area is declining, it could be (and is often assumed) that the sea ice concentration is declining everywhere.

- The Arctic MIZ extent is an indicator for the extent of habitat for extremely important biological activity in the Arctic. This is the first study that provides this metric/indicator. While width might be a proxy for extent, it becomes an indirect indicator of extent due to the retracting northward movement of the MIZ.

- Examples of biological activity dependent on the extent of the marginal ice zone have been added in Section 5.3, starting at line 355. Please see also the response to Specific Comment #1 from Reviewer #3.

- Because the MIZ has been shown to be important also in the physical Arctic climate, the timeseries of the extent metric for the MIZ is interesting for a wide variety of Arctic fields of study.

2) The errors in the sea ice concentration retrievals from passive microwave satellites during summer are large. For example, in their figure 3 they show wildly varying estimates of where the northern edge of the MIZ is. Some (Walt Meier and/or others at NSIDC or NASA may have a paper on this) have estimated the summer SIC error to be higher than 40% during summer, and most of this error and differences between the retrieval methods is related to how they filter weather. Rolph et al. need to provide a more thorough error analysis than assuming an overall 10% error estimate since the errors in the SIC retrievals affect how robust their conclusions are.

- We agree and show in our results that the generally applied 10% error is for retrieval of sea ice concentration is not valid for the summer period. Indeed, our analysis demonstrates that MIZ quantities based on current sea ice concentration retrievals are not accurate enough to constrain model results. To avoid misinterpretation, we have added to the manuscript in the Discussion Section 5.5 at lines 381-382: 'It is clear from the differences in the observations that the uncertainty varies seasonally and often exceeds 10%, with the greatest uncertainty in August (Figures 2 and 3).'

- Increasing the uncertainty of the sea ice concentration will not change the main result of the paper that the MIZ is not exhibiting a significant trend in extent. For this reason, the robustness of the conclusion still stands without increasing the error in summer to 40% for example.

- Spreen et al (2008) gave an error between 10-12% between the sea ice concentration observations from a summer expedition with the German icebreaker Polarstern and three separate algorithms used to process AMSR-E satellite data.

3) The fact that models don't reproduce these observations isn't surprising. There are already many papers that show that various models don't reproduce some observation. But as with any tool, does simply showing that a tool doesn't work for this job warrant publication? If Rolph et al. could pin down what needs to be improved in the models, that would advance science and the inclusion of the model study would be interesting.

- We included the model experiment in order to understand how the MIZ extent, as calculated from satellite-derived sea ice concentrations, compares with the range of MIZ extent as calculated from the model results. We found that modelled MIZ extent does lie within the range/uncertainty of the observations (please see dashed lines in Figure 1).

- Please note we do not conclude that the model does not reproduce observations, but that the observations of the MIZ are not accurate enough to constrain model results.

Minor Suggestions:
4) Be consistent in your use of units. E.g. in lines 194-195 you switch between meters squared to kilometers squared. I suggest sticking with kilometers squared.

Yes, this has been changed now to kilometers squared.

5) Need to note 10^7 in the label for the Y axes in Fig. 1 rather than "1e7" on the top corner of the plots.

Yes, thanks, this has been changed to $10^7$ in Figure 1.

6) Provide a short section 3.3 discussing how statistical significance was estimated. Maybe just move this from caption of table 1.

We have moved this from the caption of Table 1 to the end of Section 3.2.

7) Caption of Fig. 1: Change "…is defined at…" to "…is defined as…".

Thanks, changed.

---

## Author Comment (AC2) · 26 Mar 2020

Authors responses in blue.

The authors present an analysis of historical MIZ extent using available satellite products and the CICE-CPOM model. They find no historical trend in extent but an increase in the fraction of the total ice that is MIZ. MIZ extent provides an interesting perspective which is complementary to the previously published trends in MIZ position and width. Within the scope of the present study, an explanation for the lack of trend drawing on MIZ geometry and prior results could strengthen and contextualize the findings.

We would like to thank Court Strong for his thorough review of our manuscript and his helpful suggestions to improve our manuscript. Following his advice, we added time series showing the poleward movement and widening of the MIZ resulting in a constant MIZ extent due to the geometry of the earth.

**Major comments:**
1. A poleward trending and widening MIZ does not necessarily need to conserve area, so the lack of trend reported here is potentially interesting. The manuscript would be strengthened by explaining how this result follows from the magnitude and direction of changes in MIZ width and position. One could, for example, simplify the geometry by approximating the MIZ as an annulus and then plug in the latitude rate of change (as a radius) and width rate of change from Table 1 of Strong and Rigor 2013). Over the satellite record, this gives changes in warm-season MIZ extent which are small relative to interannual variability.

   • Simplifying the MIZ shape to an annulus presented problems because we found that certain months (especially March) had pack or landfast ice south of the MIZ, and so it was difficult to determine true MIZ area in this way. Instead, we approximated the MIZ area by first finding the average of latitudes over all the grid cells that were defined as MIZ. Using this latitude and assuming a spherical earth and no land, we found the average MIZ perimeter. Because we assumed no land when calculating the average perimeter of the MIZ, we focused on the months when the ice is, in general, north of the main northern hemisphere landmass. Following this, further analysis of the summer months (which show the most change in relative MIZ fraction) is shown below. The changes in average MIZ latitude and MIZ width are shown in Figures 1 and 2, respectively.
   • Since we had previously found the extent of the MIZ (Figure 1 in the manuscript), the MIZ width could be found from Width = Extent / Perimeter.
   • For each month, the change in width and change in perimeter were both calculated from the slope of each yearly timeseries. These methods have been added as a new section in the manuscript (Section 3.3)

**Figure 1. Timeseries of average MIZ latitude**

[Figure]

**Figure 2. Timeseries of MIZ width**

[Figure]

Table 1. Trends of MIZ latitude and width change based on monthly means of sea ice concentration for July, August, and September. Only significant trends at a 95% confidence level are shown. RMS values of the detrended timeseries are given in parenthesis. Timeseries of latitudes and widths from where these trends originate for Bootstrap (black), OSI-450 (blue), and CICE-CPOM-2019 (red) are shown in Figures 1 and 2 respectively. AMSR timeseries were excluded due to the limited number of years in those datasets.

| | July | | | August | | | September | | |
|---|---|---|---|---|---|---|---|---|---|
| **ΔMIZ latitude [deg/year]** | **Bootstrap** | **OSI-450** | **CICE-CPOM-2019** | 0.068 | **0.065** | **0.122** | 0.074 | **0.069** | **0.159** |
| | 0.039 | **0.036** | **0.069** | | | | | | |
| **RMS for ΔMIZ latitude** | 0.387 | **0.484** | **0.806** | 0.607 | **0.667** | **0.998** | 0.708 | **0.896** | **1.13** |
| **ΔMIZ width [km/year] (RMS)** | 0.720 | **Insignif.** | **6.50** | 1.11 | **2.19** | **4.06** | 0.55 | **Insignif.** | **Insigif.** |
| **RMS for ΔMIZ width** | 18.3 | **--** | **59.7** | 26.3 | **59.6** | **96.9** | 17.5 | **--** | **--** |

- We show, in agreement with Strong and Rigor (2013), that the interannual variability (RMS values in Table 1 above) of both the mean latitude of the MIZ and the mean width is roughly 10 to 30 times larger than annual trends. Since the MIZ extent is a function of latitude and perimeter, it also shows that the change in MIZ extent is small relative to interannual variability.

- We have summarized the latitude trends given above in Table 1 to the Results Section 4.3, starting at line 267.

- We also compared these changes of the MIZ width and latitude calculated from the Bootstrap, OSI-450, and CICE-CPOM-2019 model output with the values of the MIZ width and latitude changes found in Table 1 of Strong and Rigor (2013). The average latitude change in the observational datasets (Bootstrap and OSI-450) agree well with the results from Strong and Rigor (2013), as seen in the bottom rows of Table 2 below (0.0603, 0.0564, and 0.059 degrees/year respectively). The model overestimates the latitude change at 0.117 degrees/year. This has been added to the Results Section 4.3, starting at line 261.

- Compared to the 1.3 km/year trend in MIZ width as found in Strong and Rigor (2013), Bootstrap shows a lower trend (0.793 km/year), OSI-450 a comparable trend at 1.49 km/year, and the model has a much higher trend at 3.72 km/year (Table 2). It should be noted that the datasets cover different temporal ranges, with the Strong and Rigor from 1979-2011 and the other datasets covering through 2017, 2015, and 2016 for Bootstrap, OSI-450, and CICE-CPOM-2019 respectively. The OSI-450 trends in MIZ width and latitude are closer to that of Strong and Rigor (2013), compared to the NASA Bootstrap. This can be attributed in part to the differences in the Bootstrap and OSI-450 algorithms.

Table 2. Comparison of MIZ width and latitude change with Strong and Rigor (2013). Only significant trends (95% confidence level) are shown for Bootstrap, OSI-450, and model data.

| | July- Sept | | | July – Sept from Strong and Rigor (2013) |
|---|---|---|---|---|
| Average width change [km/year] | Bootstrap (1979-2017) | OSI-450 (1979 – 2015) | CICE-CPOM-2019 (1979-2016) | (1979-2011) 1.3 |
| | 0.793 | 1.49 | 3.72 | |
| Average latitude change [deg per year] | 0.0603 | 0.0564 | 0.117 | 0.059 |

2. Related to above, the authors touch on the concept of perimeter briefly in their remarks on lines 1 and 260, but this can be made more quantitative and also contextualized by prior related work. For example, Strong et al. (2017) calculated pan-Arctic MIZ extent in the bootstrap data, denoted by A in their equation (15), and used this time series in conjunction with MIZ perimeter (L) to study the width trend. They also concluded that the widening is consistent with the decline in the inner pack ice area outpacing the decline in total ice area (expressed as effective radii; trends reported at the end of their Section 4a and Fig 8b).

- This is a good suggestion, and we now have quantitatively compared the necessary changes in width for the MIZ extent to remain constant.
- We calculated how much the MIZ width needs to change in order to keep its area constant, using the equation Area = Perimeter * Width, and set $dA/dt = 0$. The trend of the latitude was used to find the fraction change of the perimeter. The approximated perimeter of the MIZ ($P_{MIZ}$) using the average latitude of the MIZ ($\Theta_{MIZ}$) is found with the following steps, where $\theta_{initial}$ is the initial latitude taken from the trendline and $\theta_{final}$ is the final latitude taken from the trendline.

- $R_{MIZ} = R_{Earth} * \cos(\Theta_{MIZ})$
- Plugging this radius into the perimeter equation for a circle:
  $P_{MIZ} = 2\pi * R_{earth} * \cos(\Theta)$
- Finding the fraction of how much the MIZ extent is reduced if the MIZ was only moving northward with no change in width can be approximated by:

$$\frac{P_{MIZ\ (final)}}{P_{MIZ\ (initial)}} = \frac{2\pi\ R_{MIZ\ (final)}}{2\pi\ R_{MIZ\ (initial)}}$$

$$= \frac{2\pi\ R_{Earth} * \cos(\theta_{final})}{2\pi\ R_{Earth} * \cos(\theta_{initial})} = \frac{\cos(\theta_{final})}{\cos(\theta_{initial})}$$

- The above gives the fraction that the MIZ extent has decreased due to the decreased perimeter from the MIZ moving northwards. Since the MIZ area remains constant (as we have shown in the manuscript), the width must increase by the inverse of the above fraction, or:

$$\text{Fraction that MIZ width must increase for area to remain constant} = \frac{\cos(\theta_{initial})}{\cos(\theta_{final})}$$

These results are given in the first row of Table 3. The second row of Table 3 compares the fraction change of the MIZ width as given from the trends calculated from the sea ice concentration data (Figure 2 above). With the exception of the model and given the simplifications of our MIZ geometry, the fractions are relatively consistent in that they support the MIZ is widening enough to keep the area constant as the MIZ trends northwards. This point has been added in a new subsection in the Discussion (Section 5.4).

- The methods described here have also been added in a new Methods subsection (Section 3.3)

- The results of Table 3 below have been summarized in the Results (Section 4.3, renamed to 'Changes in MIZ location and geometry') starting at line 368. Table 3 has also been also added to the manuscript.

Table 3. Fraction changes of MIZ width needed for the MIZ area to remain constant compared with calculated trends in MIZ width assuming an averaged perimeter

| | July | | | August | | | September | | |
|---|---|---|---|---|---|---|---|---|---|
| Required fraction change of MIZ width for MIZ area to remain constant | Bootstrap | OSI-450 | CICE-CPOM-2019 | 1.20 | 1.20 | 1.56 | 1.23 | 1.21 | 1.75 |
| | 1.10 | 1.09 | 1.20 | | | | | | |
| Calculated fraction change from MIZ width trends | 1.16 | Insig. | 2.42 | 1.24 | 1.25 | 1.49 | 1.17 | Insig. | Insig. |

- The widening trend found in Section 4a of Strong et al. (2017) with the $l_{per}$ definition is 40% for the period of 1979-2015 in July through September. This is slightly more than the 37% widening in the 1979-2012 period as reported in Strong and Rigor (2013). Our data show lower widening trends (order of 20%, Table 3) but we think still roughly comparable given the simplifications used in the above approach compared to those methods in Strong et al. (2017). This has been added to the new Discussion Subsection 5.4 'Increase in width compensates for decrease in perimeter'.
- We have also added a statement in the Discussion Section 5.3 of the revised manuscript at lines 339-340 that the inner pack ice is outpacing the decline in total ice area with the reference to Strong et al. (2017).

3. Section 3.1: For model validation, the interpolation of concentration onto the model grid makes sense. However, to provide a definitive statement on MIZ extent trends, why not use the native 25-km NSIDC grid? I think the nominal resolution around the pole in the 1-degree tripolar grid is about 85 km, although line 100 in Section 2.2. mentions _40 km. Either way, potential artifacts of the regridding and interpolation should be considered because MIZ width ranges from about 50 to 150 km.

- The nominal resolution of our 1 degree tripolar grid is 40-km in the Arctic (as stated in the manuscript). The regridding from a 25-km grid to our 40-km grid has no significant impact.
- We have shown in our response to Comment #1 (please see Table 2 above) that our latitude trend data is consistent with that of Strong and Rigor (2013)

4. The abstract states that the MIZ is "trending northwards" and Section 4.3 is titled "MIZ trending northwards," but the presented results seem restricted to maps of August 1993 and August 2013. I did not see the record-length analysis to support the statement in the abstract "The MIZ is trending northwards, consistent with other studies" (line14).

- Yes, we agree with the reviewer's comment, and that the quantified latitude and width trends add support to this statement. Please also refer to our answer to Comment #1.
- We have added the figure showing the timeseries of MIZ latitudes as Figure 3 to the revised manuscript and have added a description of the latitude trends given above to the Results Section 4.3, starting at line 260. We have also added a reference to the Figure 3 in Discussion Section 5.3 at line 355.

5. The MIZ fraction change is reported as "small" in the abstract, and a quantitative value would be informative here. Also, is it really small? If I understand the units correctly, a 0.003 / year trend would amount to an increase of 0.117 MIZ fraction over the record. For a quantity starting round 0.2, increasing to 0.3 would be a 50% increase.

- Yes, this is a great point, and for all of the datasets, the change has now been calculated in terms of % increase, in addition to the previously stated fraction per year units. A column was added to Table 1 of the revised manuscript, and is also shown below.
- The statement in the abstract (starting at line 16) that had indicated the relative MIZ change is small has now been changed to the following: 'We find a large and significant increase ( >50%) in the August and September MIZ fraction (MIZ extent divided by sea ice extent) for the Bootstrap and OSI-450 observational datasets, which can be attributed to the reduction in total sea ice extent.'

Table 1. Added column of % increase of MIZ cover compared to total ice extent. Other columns are trends of total ice extent, MIZ extent, and extent of MIZ relative to total ice extent.

$\alpha = 0.05$

| Trend in $10^{10}$ m$^2$ per year (r$^2$) | | MIZ extent | Relative MIZ (MIZ extent/total ice extent) [1/year] | Total change Relative MIZ (%) |
|---|---|---|---|---|
| **March** | Total ice extent | | | |
| OSI 450 | -2.42 (0.74) | Insig. | Insig. | Insig.* |
| Bootstrap | -2.76 (0.78) | -0.520 | Insig. | Insig.* |
| AMSR | -3.04 (0.43)* | Insig.* | Insig.* | Insig.* |
| CICE-CPOM-2019 | Insig. | Insig. | Insig. | Insig.* |
| **July** | | | | |
| OSI 450 | -5.27 (0.84) | Insig. | +0.003 (0.375) | 27% |
| Bootstrap | -5.85 (0.87) | Insig. | +0.002 (0.450) | 38% |
| AMSR | -7.55 (0.67)* | Insig.* | Insig.* | Insig.* |
| CICE-CPOM-2019 | -4.29 (0.70) | Insig. | +0.009 (0.636) | 124% |
| **August** | | | | |
| OSI 450 | -6.52 (0.78) | Insig. | +0.005 (0.479) | 50% |
| Bootstrap | -7.19 (0.81) | Insig. | +0.003 (0.444) | 56% |
| AMSR | -7.96 (0.47)* | Insig.* | +0.008 (0.672)* | 60% |
| CICE-CPOM-2019 | -9.61 (0.71) | Insig. | +0.010 (0.557) | 91% |
| **September** | | | | |
| OSI 450 | -7.80 (0.75) | Insig. | +0.004 (0.392) | 79% |
| Bootstrap | -8.07 (0.75) | Insig. | +0.003 (0.479) | 66% |
| AMSR | -9.72 (0.50)* | Insig.* | Insig.* | Insig.* |
| CICE-CPOM-2019 | -9.02 (0.79) | -1.37 (0.31) | +0.003 (0.293) | 57% |

- We have also amended a similar statement in the conclusion so that it now reads (starting at line 397): 'Due to the decrease in Arctic sea ice extent, there is a significant increase (> 50%) in the relative MIZ extent (MIZ extent divided by sea ice extent) during August and September for the Bootstrap and OSI-450 observational datasets. During July and August, the positive trend is 2 to 4 times stronger in our model simulation than these observations.'

6. We see that the model performance varies through the year as discussed in Section 4.1, but it is difficult to interpret the discrepancy from the warm-season observations because the spatial pattern is left implicit. Does the total extent error signal that the model MIZ has a position error, width error, or both? A more spatially explicit treatment of the model performance would help the reader to understand the purpose of including the model, and its intended role and weight in the suite of results.

- We have included a model to examine the extent to which the observed changes could constrain models and the extent to which the model represents observations. But this can only be indicative without a much larger study. This is particularly interesting in considerations of future projections of changes of the MIZ (e.g. Aksenov et al, 2017).
- Figure 5 in the revised manuscript (previously Figure 3) gives an indication of the spatial discrepancy between the model and the observations. This is especially true during the summer months.
- Our primary purpose is to examine changes in the observed marginal ice zone, and we have shown that any further analysis of the spatial patterns of MIZ in model output will be very poorly constrained by the observations. We have added a statement in Section 4.3, lines 284-285 to make this more clear: 'The spatial variability of the MIZ is poorly constrained by observations' with a reference to Figure 5.

7. Suggest including a paragraph somewhere in main text to detail the statistical methods (assumed degrees of freedom, tests were parametric versus bootstrap, etc.).

- The following statement regarding the statistical method has been removed from the caption of Table 1 and added at the end of Section 3.2: ''A linear least-squares regression was used to calculate the trends, using a 95% confidence level.''

8. The title is very general. To more precisely reflect the presented analysis, suggest something like: Historical analysis of Arctic marginal ice zone extent''.

- Agreed, the title has been changed to 'Changes of the marginal ice zone during the satellite era.'

**Minor comments:**

1. Line 11 in abstract: I did not see an extrapolation of the results forward in time in the paper. If this remark just follows from the report of no trend, suggest removing to avoid implying that a supporting extrapolation with uncertainty analysis was performed.

Yes, this statement has been removed.

2. Lines 14-16 recommends that future authors "provide a specific and clear definition when stating that the MIZ is rapidly changing." Suggest an edit here to clarify if future authors are being asked to specify the MIZ definition or to specify the particular MIZ property that is changing (width, area, latitude, etc.).

The sentence has been changed to 'Given the results of this study, we suggest that references to 'rapid changes' in the MIZ should remain cautious and provide a specific and clear definition of both the MIZ itself and also the property of the MIZ that is changing.'

3. Lines 22-24 state that the cited studies "tend to assume that marginal ice zone (MIZ) extent is increasing." I am familiar with these studies and looking back through a few of them as a sample, found no assumption that MIZ extent is increasing. Instead, the remarks about MIZ change were literature-based and referred to specific properties.

Some statements in the above references that gave the authors this impression that the MIZ is increasing in extent are listed below. We realize that other specific properties are what might have been referred to here, but one of the suggestions of this work is to clearly indicate which property of the MIZ is expanding. This way, to say 'the MIZ is expanding' will not be interpreted as the MIZ extent is increasing (which, as we have presented in this paper, is not what the satellite data show).

- 'The Arctic Marginal Ice Zone … is expanding as the result of on-going sea ice retreat ' (The first statement of the abstract in Boutin et al (2019). )
- A new reference (under review) has also been added to this line, which states in the first statement of the abstract: 'The decrease in Arctic sea ice extent is associated with an increase of the area where sea ice and open ocean interact, commonly referred to as the Marginal Ice Zone (MIZ).' (Boutin et al 2020a).

- 'The most dramatic intra-annual variability in sea-ice cover is found in the MIZ … As summer sea-ice cover becomes thinner and more fractured, these regions will become larger'. (A statement in the introduction of Horvat and Tziperman (2015))
- 'Summertime opening of the Beaufort and Chukchi Seas has amplified the extent … of the seasonal MIZ, the region of fractional ice cover that forms the transition between open water and pack ice' (Lee and Thomson 2017).
- 'These changes in Arctic sea ice extent suggest scientifically important changes in the position, width, and area of the marginal ice zone' (Strong et al 2017) Width and position are also referred to in this statement, but this study shows that area should not be assumed to also change.
- After searching again through the other references in this statement, the authors removed those references where this assumption couldn't be clearly identified.
- These statements appear at lines 25-26 in the Introduction.

4. Why was the NSIDC Climate Data Record not used? I think one of the motivations for CDR was to develop a consistent record suitable for trend analysis.

    Our selection of satellite products OSI-450 (EUMETSAT), NASA Bootstrap, AMSR-E and AMSR-2 provide an adequate representation. Differences between NSIDC CDR and OSI-450 are small with respect to shown discrepancies as shown in our results.

5. Line 202: It's not clear what is meant by "The interannual variability of the MIZ … varies more than the sea ice extent." A more precise statement referencing specific variance statistics could clarify.

    We have provided variance statistics of the detrended MIZ width and latitude timeseries for 3 datasets and these are given in Table 1 above. We have changed this statement (now at line 232 of the revised manuscript) so it now refers to the spread of the MIZ observations being larger than the spread of the observations for sea ice extent, especially in the summer months.

6. Line 212 and thereafter. Suggest using a consistent format when referring to the MIZ fraction trends. Something like "0.003 per year" as in the Table seems less likely to confuse than 0.3% the latter could be interpreted as a percent change rather than change in percent).

    Yes, we agree. The text starting at line 241 in Section 4.2 has been changed so the numbers match the same format as the table. (0.3% has been changed to 0.003 per year, etc) in the text.

7. Line 238: "Our results are robust" – not clear which specific results are referred to here.

    The sentence has been rephrased to 'The lack of trend in MIZ extent is robust given changes in the upper and lower bounds of the sea ice concentration in the MIZ definition'. This statement now appears at lines 290-291.

---

## Author Comment (AC3) · 26 Mar 2020

Reviewer 3

Authors' responses shown in blue.

**General comments**
The manuscript "Changes of the Arctic marginal ice zone" by R. Rolph, D. Feltham, and D. Schröder provides a clear analysis of evolution in Arctic marginal ice zone (MIZ) extent relative to total sea ice extent (SIE) in a changing climate. In highlighting, based on an operational definition, that the MIZ extent shows no significant trend over the last 40 years despite a decline and well-defined trend in total SIE, this analysis underscores the need for a universal definition for the MIZ, identification of relevant variables in addition to extent for its characterization, and improved understanding of implications in a changing climate for communities influenced by MIZ processes.

This paper addresses relevant scientific questions including characterization of the MIZ, and presents novel analysis that contributes to an understanding of changes in the sea ice cover, and in particular poleward migration in MIZ and total SIE, in the context of a changing climate.

We thank the reviewer for the helpful comments to improve the manuscript. Following suggestions from reviewer Court Strong, we added timeseries of the mean MIZ latitude (Figure 3 of the revised manuscript) and width (Figure 2 in the response to Court Strong). These illustrate a consistent picture that the northward shift compensates the widening of MIZ such that the MIZ extent remains constant with time.

Also of interest however is the sensitivity of this analysis to the mathematical and physical definition for the MIZ; investigation of additional techniques used to analyse total SIE (i.e. geographic muting described in Eisenman, 2010) applied to the MIZ that could perhaps explain the absence of statistically significant trends in MIZ extent over the past 40 years and, as noted by other reviewers; further exploration of reasons for the absence of changes in MIZ extent; in addition to alternative MIZ variables/aspects (area, regional variability, zonal mean MIZ edge as in Eisenman, 2010) that do reflect changes in the zone between fully ice-covered and ice-free regions in response to global warming. This is therefore to recommend that the manuscript be published following revisions that address MIZ definitions and analysis. Please find below more specific comments for consideration.

We note that geographic muting only applies to those months where the sea ice would extend beyond the limit of land, if the land was not present. So, during the summer months, the geographical muting would not well explain the lack of change in the MIZ. We have added a statement reflecting this point in Section 5.1, starting at line 304. As indicated above and in our response to Court Strong, our additional analyses of mean MIZ latitude and width provide extra insight into these conjoining factors involved in the evolution of the MIZ.

We note that the reviewer refers to the term 'MIZ area' above, and we have taken this to mean the sea ice area within the MIZ, given that the MIZ extent has already been calculated. As the reviewer has suggested, we have found the sea ice area within the MIZ for March, July, August and September (Figure 1 below and as an added Figure 4 in the revised manuscript). We found no significant trends of the sea ice area within the MIZ in March except for a slight negative trend for the Bootstrap dataset (-0.0025 x $10^6$ km$^2$/year). In July, there is a significant positive trend for the model at 0.027 x $10^6$ km$^2$/year and in September, a slight negative trend for the model at -0.0092 km$^2$/year. The other datasets showed no significant

trend in sea ice area within the MIZ. This has been added to the Results Section 4.3, starting at line 280.

Due to the clearly large inconsistencies in the observations in the regional location of MIZ (please see Figure 5 in the revised manuscript), analyses of the regional trends and locations of the MIZ do not give much indication of the regional trends in reality. Until the observations of the sea ice concentration are improved and the observational datasets agree more with each other in both spatial and temporal variability, a regional trend analysis would give unrealistic (or impossible to validate) results. We have added a statement in the Results Section 4.3, line 284: 'The spatial variability of the MIZ is poorly constrained by observations.'

Figure 1. Sea ice area within the MIZ. Monthly averaged from daily data.

[Figure]

**Specific comments**

*Abstract*
p. 1, lines 6 – 8. "It does not logically follow, however, that the extent of the marginal ice zone (MIZ), here defined as the area of the ocean with ice concentrations from 15 to 80%, is also changing". What are the implications of assumptions associated with a changing MIZ extent?

Some implications of assumptions associated with MIZ extent are:
- If one were to assume the MIZ extent is changing, we may be focusing on the wrong aspect (e.g. instead of the MIZ moving northward and widening) with regard to change in other parts of the climate system (e.g. phytoplankton populations).
- A changing MIZ extent would have implications for the level of atmosphere and ocean mixing within the ice-covered region, e.g. if the MIZ extent were to increase, we

would likely see an increase in the heat flux between the ocean and atmosphere in these partially ice-covered regions.

- An increase in MIZ extent could increase the level of gas exchange and could have consequences for the amount of greenhouse gases absorbed and released by those regions of the ocean containing sea ice.
- MIZ extent is a metric for the area of vital habitat for important Arctic biological life and also for Arctic primary productivity. A change in the MIZ extent would result in further changes to the extent of this habitat.  For example, ice algae grow on the underside of (and within) the sea ice and are an early important food source for zooplankton and ice fauna (Horner et al. 1992; Hegseth, 1998; Søreide et al., 2013). The deformed ice in the MIZ creates ridged habitats underwater for animals such as polar cod (Hop and Gjøsæter, 2013) and also habitats above the sea ice for animals such as seals, polar bears, and seabirds (Hamilton et al., 2017).   These statements along with the references have been added at the end of Discussion Section 5.2.

We have added a statement to the Abstract lines 7 – 9: 'Changes in the MIZ extent has implications for the level of atmospheric and ocean heat and gas exchange in the area of partially ice-covered ocean, as well as for the extent of habitat for organisms that rely on the MIZ, from primary producers like sea ice algae to seals and birds.'

p.1, lines 14-16. "Given the results of this study, we suggest that future studies need to remain cautious and provide a specific and clear definition when stating the MIZ is 'rapidly changing'." Perhaps provide an appropriate definition and context for the statement of a 'rapidly changing' MIZ. As is noted below, additional MIZ definitions and changes in additional MIZ characteristics over the past 40 years could be evaluated and compared with MIZ extent to determine whether these properties and attributes capture a rapidly changing MIZ.

- The statement has been changed (also taking into consideration the comment from Reviewer #2) to: 'Given the results of this study, we suggest that references to 'rapid changes' in the MIZ should remain cautious and provide a specific and clear definition of both the MIZ itself and also the property of the MIZ that is changing.'
- An additional MIZ characteristic we have now evaluated is the sea ice area within the MIZ, and has been added to the manuscript.  Please see the Results section 4.3 starting at line 280.

*Introduction*

p. 2, line 45. Perhaps include 'extent' following 'MIZ'.

- This statement has 'extent' left out to suggest that future authors should define very specifically what about the MIZ is changing, whether it be extent or other properties. We have changed the sentence so it now reads as: 'Thus, we need to remain cautious and provide a specific and clear definition of the property of the MIZ when stating that 'the MIZ is rapidly changing.''  This statement is now at line 48.

p. 2, lines 45 – 46. "It also follows that we need to be aware of the extent to which our observations are able to constrain any model of the MIZ". Does this study also highlight the need for a universal and/or alternative definition for the MIZ?

- The statement here was meant to inform the reader that because there is no clear observational value of MIZ extent, any model which shows MIZ location (as defined by sea ice concentration at least) cannot be well-validated in this context through observation.
- If one were to change the definition of the MIZ such that it could then be constrained by observations, this would likely require further definitions/analysis to answer the MIZ research question involved and still presents an issue. Please see also the response below (for p.2. L57).

p. 2, line 57. "Here we also describe how we defined the MIZ and sea ice cover in our calculations". Will the results from this analysis differ for different MIZ definitions?

- Yes, we would expect that the MIZ extent would change if the MIZ definition were to change. The reason that the sea ice concentration was used is that the MIZ is readily calculable due to the fact that sea ice concentration data is available.
- Another common definition of the MIZ is that region where ocean waves can influence the ice cover, but this requires data that is not readily available on a pan-Arctic scale in comparison to sea ice concentration. There are benefits and drawbacks to the definition of the MIZ as the region of partially-ice covered ocean that is impacted by ocean waves.
- We have added a Discussion Section 5.1 starting at line 293: 'Differing definitions of MIZ extent':
  'Similar to sea ice extent, the MIZ extent is also defined by sea ice concentration thresholds. Another definition of the MIZ in common usage is that the MIZ (e.g. Squire, 2020) is that region of partially-ice covered ocean that is impacted by ocean waves. One drawback of this definition is that it necessitates further definition of where the ice-covered ocean is deemed to be 'impacted by ocean waves'. This could be problematic because different applications (e.g. shipping, climate studies) could require different thresholds of when they consider waves important. There are also significant uncertainties with both observing and forecasting waves within the sea ice and this is an ongoing field of study (Roach et al., 2019; Stopa et al., 2018). For instance, it has been shown that ocean waves can penetrate deeper into the ice pack than previously thought (Kohout et al., 2014). Although the definition of the MIZ using ocean wave penetration can be very useful for other studies (for example, boundary layer air-sea interaction or wave-action studies), we argue that comparisons of purely MIZ extent from different observational datasets and models should be done through sea ice concentration thresholds. This is especially true for model comparisons given the unknowns in wave-sea ice interaction (Squire, 2020). Some techniques used to analyse total sea ice extent such as geographical muting (Eisenman, 2010) only apply to those months where sea ice extends beyond the limit of the land, if the land was not present. During the summer months, the geographical muting would not well explain why the MIZ extent remains constant. '

p. 2, line 58. The timeframe could be indicated following "March, July, August, and September".

- Yes, thank you, agreed; the phrase 'for the period from 1979-2017' has been added after these month names. And this now appears at lines 62-63.

*Methods*

p. 6, lines 167 – 170. Perhaps the MIZ area could be examined in addition to MIZ extent, and results compared to characterize changes relative to total SIE and area over the past 40 years.

- Yes, the MIZ area (sea ice area within the MIZ) has now been calculated for all of the datasets evaluated in this manuscript, and the results are presented in Figure 1 above as well as added to the revised manuscript as a new figure (Figure 4) . Please see also the response to the 'General Comments' section).
- We have added a statement in the Methods section to include that this analysis has been done (lines 174-175, Section 3.2).
- A statement has also been added to Results Section 4.3 starting at line 280: "Although the MIZ is trending northwards, the observations do not support any trend in its overall sea ice area, with the exception of March for Bootstrap at -0.0025 x $10^6$ km$^2$ per year (Figure 4).  The modelled sea ice area within the MIZ did not show a trend except for July and September at 0.027 x $10^6$ km$^2$ per year and -0.0092 km$^2$ per year, respectively (Figure 4)."
- Given that there is a lack of trend in the sea ice area within the MIZ, consistent with the lack of trend in the MIZ extent, further comparison to the decline in the sea ice extent we feel will not give important new insights.
- In the Discussion Section 5.2, we have added these statements at line 327: However, we have found no trend in the observations of sea ice area in the MIZ except for the slight negative trend in March in the Bootstrap data, but the spread of the sea ice area within the MIZ across the observational datasets is large (Figure 4).  Due to this, there could possibly be a trend in the MIZ sea ice area which we are not able to resolve.  For example, the slight significant trends of sea ice area in the MIZ shown by the model are still within the range of observations.  Since there is no trend in sea ice area within the MIZ and no trend in the MIZ extent, there is no significant change of sea ice concentration within the MIZ based on observations. It follows that the pan-Arctic averaged sea ice concentration is not declining in concert with its declining extent. This suggests that changes to the extent of the MIZ depend strongly on the sea ice thickness distribution.
- We have also added a new statement in the Conclusions section pertaining to sea ice area, starting at line 399-400.

p. 6, lines 176 – 177. "…an error of 10%..." Does this uncertainty vary seasonally?

- Yes, this is a good point, and although we have applied an error of 10% for our observations, our results clearly show there is an uncertainty in the sea ice concentration that varies seasonally.  Although the existing literature also support that the uncertainty varies seasonally, there are no robust uncertainty values to apply to our data.
- We added a statement in the Discussion section 5.4 (lines 381-382) that states: 'It is clear from the differences in the observations that the uncertainty varies seasonally and often exceeds 10%, with the greatest uncertainty in August (Figures 2 and 3).'

p. 6, lines 177 – 178. Perhaps conduct the same analysis for sea ice area, MIZ area, and relative MIZ area.

- We have now expanded the analysis of the manuscript to include the sea ice area within the MIZ (Figure 1 above and new Figure 4 in the revised manuscript).  Since both the sea ice area within the MIZ and the MIZ extent do not show a trend, the sea

ice area within the MIZ relative to the MIZ extent will also not show a trend.  Please see also the end of the new Section 5.2.

*Results*

p. 7, line 195, and p. 8, line 230. Absence of trend in MIZ sea ice extent and northward migration in MIZ. The absence of statistically significant trends in MIZ extent suggests poleward migration of the southern and northernmost MIZ boundaries at comparable rates. Application of the zonal-mean sea ice edge concept outlined in Eisenman (2010) to the northernmost and southernmost boundaries (in a sense converse to the SIE analysis, since with a deteriorated sea ice cover the northern boundary is less stable and muting less pronounced) would illustrate rates of change for each, as well as regional variability. Also of interest is the transition to lower sea ice concentrations in the MIZ over the past 40 years, documented by MIZ area. Please see also comments pertaining to the Discussion.

- We have shown that the MIZ extent is not showing a significant trend and, since it is trending northward (causing its perimeter to shrink on a spherical earth), the MIZ must be widening. This means that the southernmost and northernmost MIZ boundaries cannot be moving northwards at the same rate.  Strong et al. (2017) found that it is the interior pack ice declining faster than the ice edge that causes the widening in summer.  This detail has now been added to Discussion Section 5.3, lines 339: 'More specifically, the inner pack ice area is outpacing the decline of total ice area, causing a widening trend (Strong et al., 2017).'
- Since we have found no robust trend in the sea ice area within the MIZ (the observations show no trend but at the same time provide room for a trend within their spread), and there is no trend in MIZ extent, it follows that the average sea ice concentration within the MIZ is not changing over the past 40 years. Please see the paragraph starting at line 326 in Section 5.2.  Please see also our response to this reviewer's first comment about the Methods section.
- Although we agree that a thorough re-analysis of the metrics presented in this paper using the Eisenman (2010) geographical-muting technique would be interesting, it would have little impact on results for the summer months where the ice is northward of land mass. Perhaps more importantly, it is difficult to interpret due to the large regional variability in the location of the MIZ (in comparison to the sea ice extent) according to the different observational products (please also see our last paragraph in the response to 'General Comments' above and the response to the comment about p. 9 L262 below).

*Discussion*

p. 9, line 256. Perhaps include the phrase 'due to decreasing total SIE' following "slightly decreasing".

This phrase has been added, and now falls at line 323 in the revised manuscript.

p. 9, line 262. Northward migration in the poleward MIZ boundary and area-weighted latitude of the MIZ. Also of interest is the study by Eisenman (2010) describing the role of zonal mean ice edge latitudes in describing asymmetry in winter and summer decline in SIE, in addition to the study by Stroeve et al. (2016) implementing a similar concept to define Antarctic MIZ boundaries according to zonal mean latitudes based also on the approach outlined in Strong and Rigor (2013). It would be interesting to see how evolution in the i) northern and ii) southern latitude MIZ boundaries/edges and iii) area (rather than extent,

based on discussions outlined in Notz; 2014) bounded by each, compares with results from the present analysis based on MIZ extent, and whether this approach captures asymmetry in the seasonal cycle as well as rates of poleward migration in the northern and southern MIZ boundaries. Evaluation of MIZ area might also illustrate the nature of transition to a lower sea ice concentration regime in the MIZ over the past 40 years.

- The responses above, and the newly introduced figures and sentences in the manuscript identified, address the evolution of the MIZ boundaries, extent, ice area within the MIZ and sea ice concentration.
- The suggestions regarding asymmetry in the summer and winter trends using the Eisenman approach are an interesting extension of our manuscript, but would be a significant undertaking out of scope of our manuscript. Moreover, we have found large discrepancies in the zonal location of the MIZ (e.g. Figures 3 and 5 of the revised manuscript) and these discrepancies would hamper a regional analysis of MIZ change in sea ice area and extent to the extent that they are unlikely to provide verifiable results.
- We have added the mean latitudes of the MIZ edge for the months of July, August, and September to the manuscript (new Figure 3 in manuscript) for the datasets Bootstrap, OSI-450, and CICE-CPOM-2019.  The mean July through September trends are significant, and the observational trends are consistent with those found in Strong and Rigor (2013) at 0.060, 0.056, and 0.059 degrees latitude per year for the Bootstrap, OSI-450, and Strong and Rigor (2013) datasets respectively.
- This trend information has been added in Results section 4.3 starting at line 261. Please see also the discussion and Table 1 of the response to Major Comment #1 by Reviewer #2 (Court Strong).
- We have added a new Methods section 3.3.  At the beginning of this new subsection, we have included statements describing how that the analysis of changes in MIZ latitude has been done.
- Also relevant is the last paragraph in our response to the 'General comments' above.

*Conclusions*
p. 10, lines 300-303. "Due to the spread of the observations in MIZ extent…" As previously noted, context for the phrase 'rapidly changing' should be provided (i.e. extent and/or other MIZ aspects including northern and southern MIZ boundaries and area).

We could not find the phrase 'rapidly changing' in the Conclusions section.  However, we do agree with the reviewer's previous comment that context for this phrase in its previous appearances should have been added to the manuscript. We have therefore added (to the last sentence of the abstract): '… definition of both the MIZ itself and also the property of the MIZ that is changing '

**Technical corrections**

p. 8, line 237. Please remove 'is'.

This have been removed.

p. 10, line 295. Perhaps replace 'big' with 'large'.

This has been replaced.

**References**

Eisenman, I., 2010: Geographic muting of changes in the Arctic sea ice cover, *Geophys. Res. Lett.*, 37, L16501, doi:10/1029/2010GL043741.

Notz, D., 2014: Sea-ice extent and its trend provide limited metrics of model performance, *The Cryosphere*, 8, 229–243, https://doi.org/10.5194/tc-8-229-2014.

Stroeve, J. C., Jenouvrier, S., Campbell, G. G., Barbraud, C., and Delord, K., 2016: Mapping and assessing variability in the Antarctic marginal ice zone, pack ice and coastal polynyas in two sea ice algorithms with implications on breeding success of snow petrels, *The Cryosphere*, 10, 1823–1843, https://doi.org/10.5194/tc-10-1823-2016.

Thank you for the opportunity to review this manuscript.

---

## Author Comment (AC4) · 26 Mar 2020

Please see the attached figure that is present in the revised manuscript as Figure 4. It is referred to in our response to the reviewer.
* * *
[Figure]

**March**

Sea ice area within MIZ [km²]

×10⁶

- Bootstrap
- OSI-450
- CICE-CPOM-2019
- AMSR-E
- AMSR-2

**July**

×10⁶

**August**

Sea ice area within MIZ [km²]

×10⁶

**September**

×10⁶

**Fig. 1.** (Figure 4 in revised manuscript) Sea ice area within the MIZ

---

## Referee Report (RR1)

The work examines satellite observations of MIZ extent, finding total MIZ extent has changed little since 1979, although/because the MIZ has moved poleward and widened. They then examine their satellite measurements against an FSD-resolving model. This is skillfull and useful analysis and the scientific content is solid.

I *am* concerned with the editorial content. I think this can be addressed without too much effort and this study would then warrant publication. My words here likely will not require near as many in response.

I was assigned as an initial reviewer, but did not upload in time. Given the three other reviewers, I will just give my main comment as I do not want to add undue burden to the authors. I hope it is useful, and I am available if the authors have questions or would like additional context.

**Main comment:** As presented, this work is motivated by arguing others: **(1)** fail to grasp the difference between a linear measure (MIZ width) and an areal measure (MIZ extent), and **(2)** use sloppy terminology like "rapidly changing".

Regarding **(1)**, you list studies that "assume MIZ extent is increasing". I wrote one! Dismayed I made such a claim without evidence, I went and checked. Here are the relevant quotes:

> . . . the Arctic marginal ice zone . . . has been widening during the summer season (*Strong and Rigor*, 2013).

and

> . . . dramatic intra-annual variability in sea-ice cover is found in the MIZ and in seasonal ice zones . . . as summer sea-ice cover becomes thinner and more fractured, these regions will become larger . . .

The MIZ is widening, and areas with seasonal ice have enlarged greatly regardless of whether the MIZ has (see *Kinnard et al.* (2008)). Having it on my desk, I do not see that *Boutin et al.* (2020) made this claim, either.

Regarding **(2)**, the MIZ *is* changing by many metrics explored here. In my reading these are often well-explained. This MS contains plenty of examples of its own loose phraseology: e.g., "significant" regional changes, correlations, trends, uncertainties, and declines, without accompanying measures of statistical significance. I don't think you need to change these, just remember that there are many hairs to split in life. I would argue that the use of "rapid" is a thin one.

Overall, I don't think you need these motivating arguments. I would remove them from the paper, leaving the focus on the presentation of MIZ, the model results, and the observation that MIZ extent is not increasing. That is a great paper to me - thanks to it, we now present MIZ extent in a recent paper (*Horvat et al.*, 2020)!

**Smaller comments:** I found that the referencing needed a careful re-examination. This was most obvious to me in papers that I am very familiar with. Therefore I would

suggest going through and making sure the referencing is accurate throughout. Examples:

- P2L33: The size-dependent melt rate problem was formulated by *Steele* (1992), not *Tsamados et al.* (2015).

- P5L137: *Horvat and Tziperman* (2015) developed the FSTD model, not an ice thickness distribution. *Roach et al.* (2018) then formulated this (with modifications) in CICE.

- P5L150: Both referenced papers do have wave spectra, but they employ different techniques. Which do you use? The exponential attenuation with floe number was first investigated by *Dumont et al.* (2011).

- P11L319: *Meylan and Bennetts* (2018) does not deal with sea ice fracture but wave scattering.

I would also point out that while this may be the first analysis of Arctic "MIZ extent", the first analysis of "MIZ extent" was probably by *Stroeve et al.* (2016).

**References**

Boutin, G., C. Lique, F. Ardhuin, C. Rousset, C. Talandier, M. Accensi, and F. Girard-Ardhuin (2020), Towards a coupled model to investigate wave–sea ice interactions in the Arctic marginal ice zone, *The Cryosphere*, *14*(2), 709–735, doi:10.5194/tc-14-709-2020.

Dumont, D., A. Kohout, and L. Bertino (2011), A wave-based model for the marginal ice zone including a floe breaking parameterization, *J. Geophys. Res.*, *116*(4), C04,001, doi:10.1029/2010JC006682.

Horvat, C., and E. Tziperman (2015), A prognostic model of the sea-ice floe size and thickness distribution, *Cryosphere*, *9*(6), 2119–2134, doi:10.5194/tc-9-2119-2015.

Horvat, C., E. Blanchard-Wrigglesworth, and A. Petty (2020), Observing waves in sea ice with ICESat-2, *Geophysical Research Letters*, doi:10.1029/2020GL087629.

Kinnard, C., C. M. Zdanowicz, R. M. Koerner, and D. A. Fisher (2008), A changing Arctic seasonal ice zone: Observations from 1870–2003 and possible oceanographic consequences, *Geophysical Research Letters*, *35*(2), L02,507, doi:10.1029/2007GL032507.

Meylan, M. H., and L. G. Bennetts (2018), Three-dimensional time-domain scattering of waves in the marginal ice zone, *Philosophical Transactions of the Royal Society A: Mathematical, Physical and Engineering Sciences*, *376*(2129), 20170,334, doi: 10.1098/rsta.2017.0334.

Roach, L. A., C. Horvat, S. M. Dean, and C. M. Bitz (2018), An emergent sea ice floe size distribution in a global coupled ocean–sea ice model, *Journal of Geophysical Research: Oceans*, *123*(6), 4322–4337, doi:10.1029/2017JC013692.

Steele, M. (1992), Sea ice melting and floe geometry in a simple ice-ocean model, *Journal of Geophysical Research: Oceans*, *97*(C11), 17,729–17,738, doi:10.1029/92JC01755.

Stroeve, J. C., S. Jenouvrier, G. G. Campbell, C. Barbraud, and K. Delord (2016), Mapping and assessing variability in the Antarctic marginal ice zone, pack ice and coastal polynyas in two sea ice algorithms with implications on breeding success of snow petrels, *Cryosphere*, *10*(4), 1823–1843, doi:10.5194/tc-10-1823-2016.

Strong, C., and I. G. Rigor (2013), Arctic marginal ice zone trending wider in summer and narrower in winter, *Geophysical Research Letters*, *40*(18), 4864–4868, doi:10.1002/grl.50928.

Tsamados, M., D. Feltham, A. Petty, D. Schroeder, and D. Flocco (2015), Processes controlling surface, bottom and lateral melt of Arctic sea ice in a state of the art sea ice model, *Philosophical Transactions of the Royal Society A: Mathematical, Physical and Engineering Sciences*, *373*(2052), 20140,167, doi:10.1098/rsta.2014.0167.

---

## Author Response (AR2)

Based on the reviewers' comments, minor changes have been made to manuscript before uploading the final version, and those minor changes are found below. Authors changes are shown in blue, or as a tracked changes version of the submitted manuscript.

**tc-2019-224** Submitted on 22 Sep 2019 **Changes of the Arctic marginal ice zone during the satellite era** Rebecca J. Rolph, Daniel L. Feltham, and David Schroeder

**Reviewer 1: Court Strong**

I appreciate the authors' detailed attention to my comments. The revised manuscript reads very well and is an excellent contribution.

My only remaining suggestions relate to clarifying the terminology associated with the quantitative analysis leading into equation (1). If I understand the calculation here, the term "radius of the MIZ" (line 190) might be more clearly described as the radius of the parallel on which the MIZ is centered (measured perpendicular to the Earth's axis of rotation). The use of the term 'MIZ perimeter' (line 191) here could be confusing if the reader envisions an annulus and considers the perimeter as the sum of the inner and outer circumferences. Perhaps replace by 'MIZ outer perimeter' or, more precisely, the circumference of the parallel on which the MIZ is centered.

Thank you, these are good suggestions, we have added two statements around lines 190:

190 RMS values. The radius of the MIZ was approximated by  $R_{MIZ} = R_{Earth} * \cos(\Theta_{MIZ})$  where  $\Theta_{MIZ}$  is the monthly-averaged MIZ latitude and  $R_{Earth}$  is the radius of the earth. The radius of the MIZ as described here refers to the radius of the parallel on which the MIZ is centered (measured perpendicular to the Earth's axis of rotation). The MIZ outer perimeter ( $P_{MIZ}$ ), or the circumference of the parallel on which the MIZ is centered, was then approximated from the average latitude of all MIZ grid cells while assuming a spherical earth and no land. This was done by substituting  $R_{MIZ}$  for the radius in the perimeter equation

**Reviewer 2: Chris Horvat**

(The main points are selected below from .pdf file he has uploaded, which is also available to the readers.)

Overall, I don't think you need these motivating arguments. I would remove them from the paper, leaving the focus on the presentation of MIZ, the model results, and the observation that MIZ extent is not increasing. That is a great paper to me - thanks to it, we now present MIZ extent in a recent paper (Horvat et al., 2020)!

We have changed the wording so that the statement now reads: '... tend to assume that the MIZ is expanding...' and have also italicized the word 'extent' in the following statement: 'The purpose of this paper is to show whether the *extent* of the MIZ, defined in this study according to the operational characterization, is actually changing.'

P2L33: The size-dependent melt rate problem was formulated by Steele (1992), not Tsamados et al. (2015).

This is a good point, and we have now added Steele (1992) before Tsamados et al. (2015).

P5L137: Horvat and Tziperman (2015) developed the FSTD model, not an ice thickness

**distribution. Roach et al. (2018) then formulated this (with modifications) in CICE.**

Good suggestions, we have changed the statement to the following:

170 floe size of 300 m, but in CICE-CPOM-2019 a joint floe-size thickness distribution (FSTD) is used which has been developed implemented and developed by Roach et al. (2018) following the ice thickness distribution of Horvat and Tziperman, (2015).

P5L150: Both referenced papers do have wave spectra, but they employ different techniques. Which do you use? The exponential attenuation with floe number was first investigated by Dumont et al. (2011).

We have removed the reference to Bennetts et al. (2017) in this line, and have left the reference (Horvat and Tziperman, 2015). This is also found in the description of attenuation in Roach et al. (2018).

P11L319: Meylan and Bennetts (2018) does not deal with sea ice fracture but wave scattering.

**We have replaced Meylan and Bennetts (2018) with Kohout et al. (2014) and Montiel and Squire (2017).**

I would also point out that while this may be the first analysis of Arctic "MIZ extent", the first analysis of "MIZ extent" was probably by Stroeve et al. (2016).

**We have added a statement in the introduction to reflect this, and it reads:**

lacking, such as quantification of the MIZ extent relative to the total sea ice extent. Thus, we need to remain cautious and provide a specific and clear definition of the property of the MIZ when stating the Arctic MIZ is 'rapidly changing'. It also

50 follows that we need to be aware of the extent to which our observations are able to constrain any model of the MIZ. Note that Stroeve et al. (2016) have examined MIZ extent in the Southern Ocean.

**Reviewer 3: Anonymous**

**General comments**

The revised manuscript "Changes of the Arctic marginal ice zone during the satellite era" by R. Rolph, D. Feltham, and D. Schröder provides a comprehensive analysis of evolution in Arctic marginal ice zone (MIZ) extent relative to total sea ice extent (SIE) in a changing climate, and clearly addresses issues noted during the review process. Evaluation of MIZ width and latitude provide additional evidence for the absence of trends in MIZ extent, while investigation of MIZ area provides further characterization for changes in the MIZ based on the operational definition of the 15% - 80% sea ice concentration threshold. Thank you for a rigorous and quantitative analysis in response to questions raised. Please find below some additional comments and questions for consideration.

1. Thank you for the figure showing changes in MIZ area for the satellite record, and for including analysis and a discussion of MIZ area. Also of interest is the change in ice concentration distributions and heterogeneity within the MIZ; this could be evaluated by examining the time series for the ratio of MIZ extent to area. In particular, does the area analysis include an evaluation of the trends in the ratio, or a comparison of trends in MIZ

**extent and MIZ area separately?**

Although the authors note in lines 337 – 343 of the revised manuscript "Since there is no trend in sea ice area within the MIZ and no trend in the MIZ extent, there is no significant change of sea ice concentration within the MIZ based on observations (where sea ice concentration in the MIZ is given as the ratio of the area of sea ice in the MIZ and the extent of the MIZ). Similarly, there would not be any trend of sea ice area within the MIZ relative to the MIZ 340 extent. Since there is also no observed change in MIZ extent, it follows that the pan-Arctic averaged sea ice concentration is not declining in concert with its declining extent. This suggests that changes to the extent of the MIZ depend strongly on the sea ice thickness distribution.", it might also be helpful to present the figure for the time series of the MIZ extent to area ratio.

We appreciate the reviewer's suggestion to add the analysis about sea ice area within the MIZ to the paper, and think this figure fits into the frame of the paper well. As we have mentioned in the last response to Reviewer #3, the time series of MIZ extent to area ratio would also not show a trend, since both the MIZ area and MIZ extent each do not show a trend. This information is given in the manuscript, so we do not think it is an essential contribution to add the suggested figure of the ratios.

2. Although trends are considered for the 1979 – 2017 timeframe, what behaviour is observed for the MIZ property (extent, area, ratio of extent to area, latitude and width) anomalies relative to the 1981-2010 climatology? Anomalies, in addition to trends, might further illustrate changes in MIZ properties in recent years.

An anomaly analysis would be interesting to include in a future study, but not essential at this point to better convey the main messages already in the manuscript.

3. Line 31. Perhaps include the phrase "defined in this study according to the operational characterization" following MIZ.

Yes, we agree and have now changed the statement to: The purpose of this paper is to show whether the extent of MIZ, defined in this study according to the operational characterization, is actually changing.

4. Figure 2. Perhaps include names of months in titles, for consistency with other figures.

Yes, thank you, the names of the months have now been added to Figure 2.

Thanks again for your responses and the opportunity to review the manuscript.

**We have also added acknowledgements to the manuscript so they now read:**

Acknowledgements: This research was funded by the National Environment Research Council of the UK (NERC), Award number NE/R000654/1. We would like thank Courtenay Strong, Chris Horvat, and 2 other anonymous reviewers for their useful comments that helped improve the manuscript. We would also like to thank the editor, John Yackel.